# Simultaneous recording of multiple cellular signaling events by frequency- and spectrally-tuned multiplexing of fluorescent probes

Michelina Kierzek[1,2], Parker E Deal[3], Evan W Miller[3,4,5], Shatanik Mukherjee[6], Dagmar Wachten[7], Arnd Baumann[8], U Benjamin Kaupp[9], Timo Strünker[1,10]*†, Christoph Brenker[1]*†

[1]Centre of Reproductive Medicine and Andrology, University of Münster, Münster, Germany; [2]CiM-IMPRS Graduate School, University of Münster, Münster, Germany; [3]Department of Chemistry, University of California, Berkeley, Berkeley, United States; [4]Department of Molecular & Cell Biology, University of California, Berkeley, Berkeley, United States; [5]Helen Wills Neuroscience Institute, University of California, Berkeley, Berkeley, United States; [6]Molecular Sensory Systems, Center of Advanced European Studies and Research, Bonn, Germany; [7]Institute of Innate Immunity, Department of Biophysical Imaging, Medical Faculty, University of Bonn, Bonn, Germany; [8]Institute of Biological Information Processing (IBI-1), Research Center Jülich, Jülich, Germany; [9]Life & Medical Sciences Institute (LIMES), University of Bonn, Bonn, Germany; [10]Cells in Motion Interfaculty Centre, University of Münster, Münster, Germany

*For correspondence:
timo.struenker@ukmuenster.de (TS);
christoph.brenker@ukmuenster.de (CB)

†These authors contributed equally to this work

Competing interest: The authors declare that no competing interests exist.

**Abstract** Fluorescent probes that change their spectral properties upon binding to small biomolecules, ions, or changes in the membrane potential ($V_m$) are invaluable tools to study cellular signaling pathways. Here, we introduce a novel technique for simultaneous recording of multiple probes at millisecond time resolution: *frequency- and spectrally-tuned multiplexing* (FAST[M]). Different from present multiplexing approaches, FAST[M] uses phase-sensitive signal detection, which renders various combinations of common probes for $V_m$ and ions accessible for multiplexing. Using kinetic stopped-flow fluorimetry, we show that FAST[M] allows simultaneous recording of rapid changes in $Ca^{2+}$, pH, $Na^+$, and $V_m$ with high sensitivity and minimal crosstalk. FAST[M] is also suited for multiplexing using single-cell microscopy and genetically encoded FRET biosensors. Moreover, FAST[M] is compatible with optochemical tools to study signaling using light. Finally, we show that the exceptional time resolution of FAST[M] also allows resolving rapid chemical reactions. Altogether, FAST[M] opens new opportunities for interrogating cellular signaling.

## Editor's evaluation

The number and temporal resolution of fluorescent probes that can be used simultaneously to interrogate cell signaling pathways are currently limited by overlap of their excitation and emission spectra. This study introduces a new technique to overcome these limitations and enable simultaneous recording of multiple probes at millisecond time resolution. The technique will facilitate studies of the temporal and causal relationships among many signaling pathways that can be targeted with fluorescent probes.

## Introduction

Cells respond to external stimuli by changes in membrane potential ($V_m$), ions, messenger molecules, or protein modification (e.g., phosphorylation or dephosphorylation). These signaling events can be monitored in real time using fluorescent probes (*Tsien, 1989*; *Rothman et al., 2005*; *Mehta and Zhang, 2011*; *Depry et al., 2013*; *Ni et al., 2018*). To delineate the network of cellular responses, it would be ideal to use different probes under identical conditions in the same sample (dubbed multiplexing) (*Keyes et al., 2021*). Such measurements can not only reveal the precise sequence of signaling events, for example, whether they are upstream or downstream of each other, but also whether events are mechanistically coupled like ion transport across membranes via exchangers or symporters (*Welch et al., 2011*; *Depry et al., 2013*). When recorded in separate experiments on different samples, inter-experimental and cell-to-cell variations may obscure temporal and mechanistic relationships of events. Moreover, by design, probes bind their target molecules, which might perturb the dynamics and sequence of cellular responses (*Lew et al., 1985*; *Haugh, 2012*; *Delvendahl et al., 2015*). Such probe-related perturbations can be inferred from multiplexing experiments.

Signaling events, such as ligand-receptor binding and changes in $V_m$ and ions, often occur on millisecond or even sub-millisecond timescales. Multiplexing of such rapid events requires kinetic techniques that allow both precisely timed stimulation of cells and simultaneous recording from different probes on a millisecond timescale. Discrimination of simultaneously excited probes relies on the spectral separation of their emissions using optical filtering (*Figure 1A*). However, the spectral space for simultaneous recording of probes is limited (*Neher and Neher, 2004*) because crosstalk arising from overlapping emission spectra compromises their discrimination. Therefore, although many spectrally distinct probes for $V_m$ and various ions and biomolecules have been developed (*Depry et al., 2013*; *Yuan et al., 2013*; *Yin et al., 2015*; *Kulkarni and Miller, 2017*; *Mehta et al., 2018*), simultaneous recording with millisecond time resolution has been restricted to two probes, for example, for two ion species or one ion species and $V_m$ (e.g., *Vogt et al., 2011*; *Jaafari et al., 2015*; *Deal et al., 2016*). For multiplexing of more than two probes, quasi-simultaneous recording has been used: probes are excited and detected sequentially by switching between different excitation wavelengths (*Figure 1B*; *Canepari et al., 2007*; *Canepari et al., 2008*; *Lee et al., 2012*; *Sulis Sato et al., 2017*; *Miyazaki et al., 2018*; *Ait Ouares et al., 2019*; *Nguyen et al., 2019*). Although quasi-simultaneous multiplexing overcomes fluorescence crosstalk, it limits the temporal resolution and, thereby, the application range for studying rapid signaling events occurring on a millisecond timescale (*van Meer et al., 2019*). Hitherto, a multiplexing strategy combining millisecond temporal resolution with high flexibility regarding the number and combinations of probes has been lacking.

Here, we introduce an approach that leverages phase-sensitive signal detection, which is commonly used to recover small signals buried in large noise (*Meade, 1983*), but also facilitates signal multiplexing (*Aslund and Carlsson, 1993*; *Carlsson et al., 1994*; *Lewis et al., 2005*; *Hwang et al., 2015*; *Garbacik et al., 2018*; *Gómez-García et al., 2018*; *Tovar et al., 2019*). We dubbed this method *frequency- and spectrally-tuned multiplexing* (FAST[M]). In brief, like conventional multiplexing, FAST[M] also involves the simultaneous excitation of different probes; however, the excitation light is modulated at distinct frequencies. The frequency-tagging of fluorescence combined with optical filtering allows discriminating probes based on their excitation and/or emission spectra (*Figure 1C*). We tested the time resolution and applicability of FAST[M] on signaling pathways of sperm and in single cultured cells. FAST[M] enabled multiplexing of at least three rapid signaling events at millisecond time resolution using various combinations of common non-ratiometric and ratiometric probes for ions and $V_m$ as well as FRET-based biosensors. Moreover, FAST[M] can be combined with kinetic rapid-mixing techniques and flash-induced release of caged messengers, for example, cGMP, to instantaneously activate signaling pathways. Finally, FAST[M] is also suited to resolve rapid chemical reactions. These unique features of FAST[M] expand the scope of time-resolved multiplexing of cellular signaling.

## Results

### Multiplexing of rapid ionic and electrical signaling events using FAST[M]

Chemosensory signaling in the flagellum of sea urchin sperm involves rapid changes in cellular messengers, ions, and $V_m$ (*Figure 1D and E*) (reviewed in: *Darszon et al., 2008*; *Strünker et al., 2015*; *Wachten et al., 2017*; *Darszon et al., 2020*); therefore, sperm are an ideal model to develop and

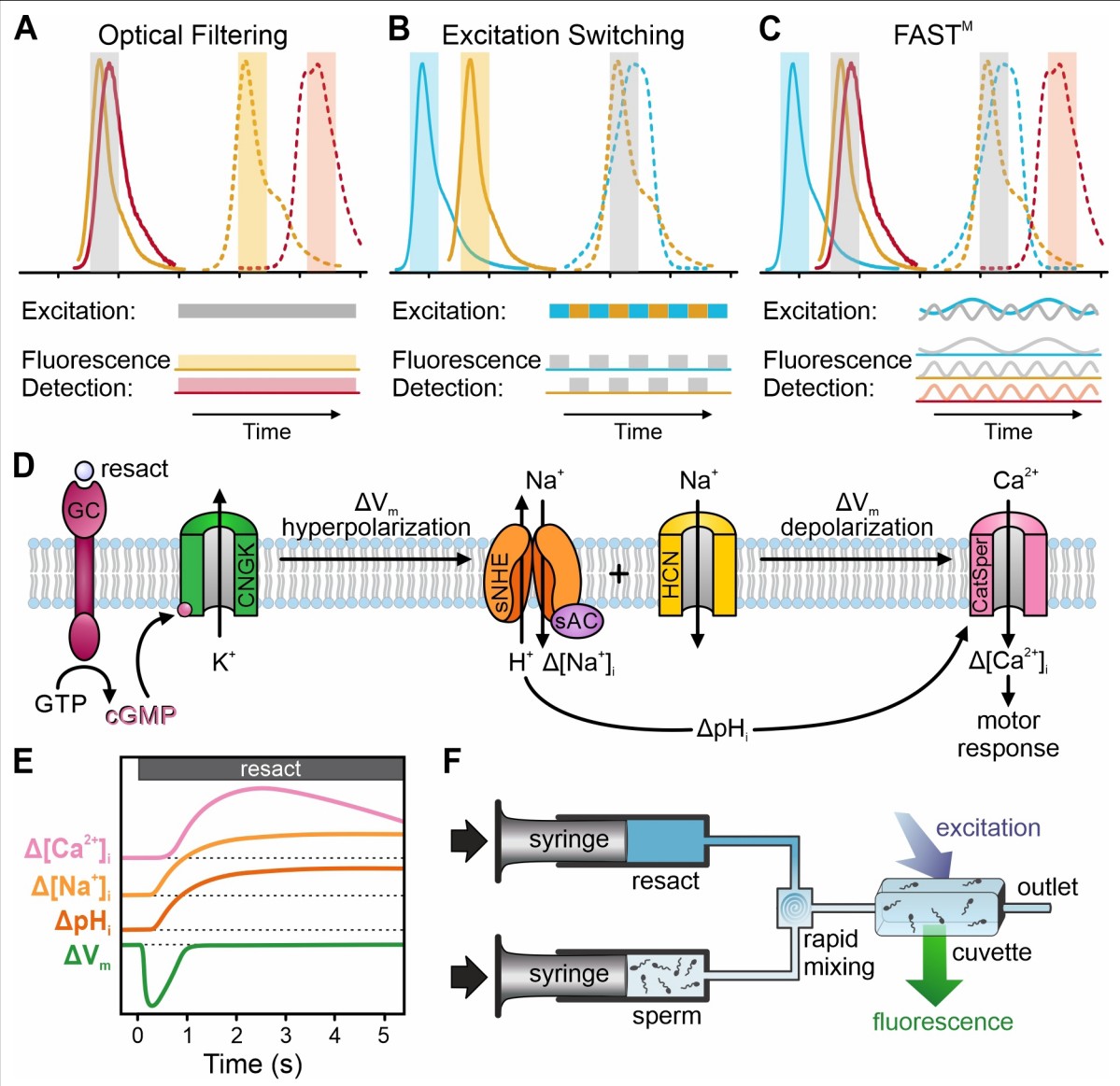

**Figure 1.** Strategies for multiplexing of fluorescent probes, and outline of the chemosensory signal transduction in the flagellum of sea urchin sperm. (**A**) Spectrally separable emission spectra (dashed) of probes allow their simultaneous recording using optical filtering. (**B**) Spectrally separable excitation spectra (outlined) allow quasi-simultaneous recording of probes using excitation-switching. (**C**) Frequency-tagging and phase-sensitive detection of fluorescence combined with optical filtering using frequency- and spectrally-tuned multiplexing (FAST$^M$) allow simultaneous recording of probes based on separable excitation and/or emission spectra. (**D**) Schematic of the chemosensory signaling pathway and (**E**) illustration of the time course of the signaling events in sea urchin sperm (reviewed in *Strünker et al., 2015*). Resact, the chemoattractant peptide released by the egg, triggers the synthesis of cGMP by activating a receptor guanylyl cyclase (GC). The rise in cGMP elicits a pulse-like $V_m$ hyperpolarization mediated by a cyclic nucleotide-gated $K^+$ channel (CNGK). The hyperpolarization activates a voltage-gated $Na^+/H^+$ exchanger (sNHE) and a hyperpolarization-activated and cyclic nucleotide-gated (HCN) channel. The $Na^+/H^+$ exchange increases $[Na^+]_i$ and $pH_i$. In turn, the increase in $pH_i$ primes $pH_i$-controlled CatSper $Ca^{2+}$ channels to open during the recovery from hyperpolarization driven by HCN channels. The resulting $Ca^{2+}$ influx drives chemotactic steering towards the egg. (**F**) Schematic of the stopped-flow setup: one syringe is filled with a suspension of probe-loaded sperm, and a second syringe is filled with a solution of resact. The syringe pistons move synchronously to rapidly mix sperm with resact in a micromixer and subsequently push this mixture into an observation cuvette, where spectroscopic measurements are performed (see *Hamzeh et al., 2019*).

test novel strategies for multiplexing. In brief, a chemoattractant peptide, *resact*, activates a receptor guanylyl cyclase. The ensuing rise of cGMP elicits a brief transient hyperpolarization, followed by an increase of the intracellular pH ($pH_i$) and $Na^+$ concentration ($[Na^+]_i$) that, ultimately, trigger a $Ca^{2+}$ influx and rise of the intracellular $Ca^{2+}$ concentration ($[Ca^{2+}]_i$) (*Figure 1E*). The sequence of signaling events

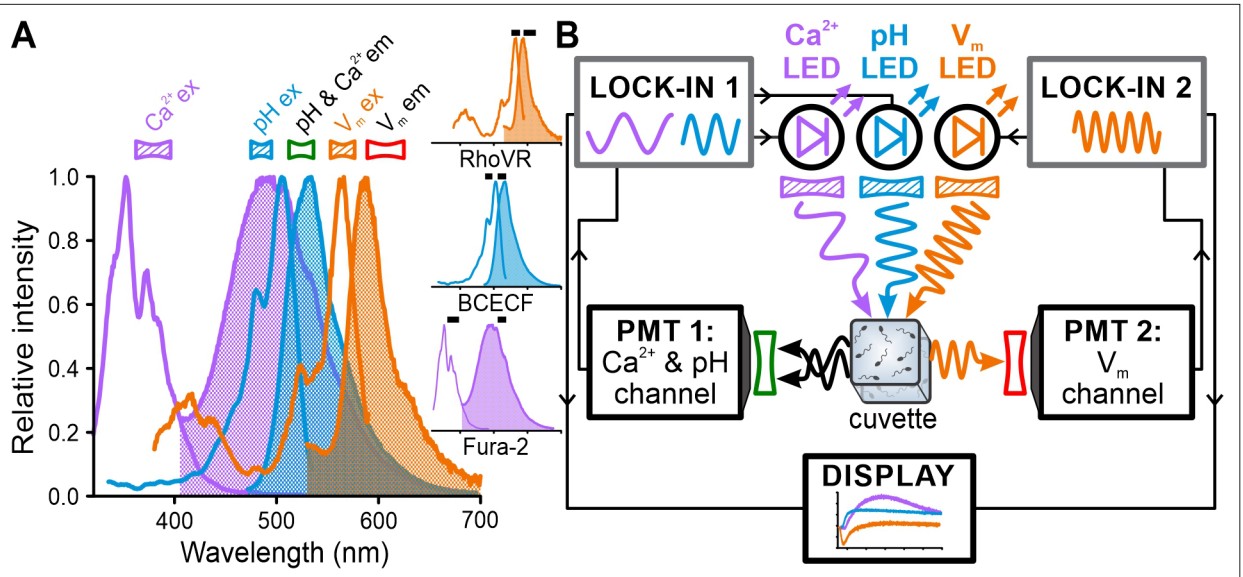

**Figure 2.** Experimental configuration for frequency- and spectrally-tuned multiplexing (FAST$^M$) of Fura-2, BCECF, and RhoVR in a stopped-flow device. (**A**) Superposition of excitation (outlined) and emission (filled) spectra of fluorescent probes for $Ca^{2+}$ (Fura-2), pH (BCECF), and $V_m$ (RhoVR). Bandpass filters used for excitation (filled) and emission (outlined) are shown above the spectra. Inset: excitation and emission spectra depicted individually with respective filters (black bars). (**B**) Schematic of FAST$^M$: each probe is excited by an LED modulated at a different frequency. The modulated emission is optically filtered and collected by two photomultipliers (PMTs). The PMT signals are demodulated by lock-in amplifiers in a phase-sensitive fashion to recover in real time $[Ca^{2+}]_i$, $pH_i$, and $V_m$ signals.

has been delineated by sequentially recording changes in either $[Ca^{2+}]_i$, $pH_i$, $[Na^+]_i$, or $V_m$ on different sperm samples using stopped-flow fluorimetry (**Figure 1F**; **Hamzeh et al., 2019**).

We set out to record the resact-induced $[Ca^{2+}]_i$, $pH_i$, and $V_m$ signals in the same sperm sample by multiplexing of the respective fluorescent probes Fura-2, BCECF, and RhoVR (**Deal et al., 2016**). The well-separated excitation spectra (**Figure 2A**) render these three probes accessible for quasi-simultaneous recording. The $[Ca^{2+}]_i$, $pH_i$, and $V_m$ signals occur, however, on a millisecond timescale, which requires their simultaneous recording; yet, due to the overlapping emission spectra (**Figure 2A**), simultaneous recording of these three probes using optical filtering alone seems intractable. Therefore, we chose to multiplex Fura-2, BCECF, and RhoVR based on simultaneous excitation by three LEDs each modulated at a distinct frequency in the kHz range (**Figure 2B**, Table 1); thereby, the emission of each probe is tagged with a unique frequency signature for discrimination. The fluorescence was collected on opposite sides of the cuvette by two photomultipliers (PMTs) equipped with appropriate optical filters: one PMT detected the emission of Fura-2 and BCECF and the other that of RhoVR (**Figure 2B**, Table 2). Lock-in amplifiers demodulated and amplified the PMT signals in a phase-sensitive fashion to discriminate, in real time, the probes based on their modulation frequencies. We refer to this approach as FAST$^M$.

We tested whether FAST$^M$ permits simultaneous recording of the three probes. First, using sperm that had been loaded with one probe only, we compared crosstalk between all three recording 'channels,' with (**Figure 3**, colored traces, FAST$^M$) and without (**Figure 3**, gray traces, optical filtering) modulating the LEDs at different frequencies. In BCECF-loaded sperm, relying on optical filtering alone, the basal fluorescence intensity ($F_o$) recorded in the BCECF channel and the Fura-2 channel (**Figure 3A**, gray) and the relative increase ($\Delta F/F_o$), reflecting the $pH_i$ response (**Figure 3B**, gray), were similar. Of note, in **Figure 3A**, the gray (optical filtering) and blue traces (FAST$^M$) in the BCECF channel are superimposed. Unsurprisingly, the BCECF fluorescence detected in the BCECF and the Fura-2 channel was similar, considering that both were collected by the same detector and optical filter (**Figure 2B**, Table 2). Basal BCECF fluorescence and the resact-induced relative increase were also detected in the RhoVR channel (**Figure 3A and B**), demonstrating that optical filtering is not sufficient to isolate the RhoVR channel from BCECF's broad emission spectrum. To quantify the crosstalk between channels, we plotted the first two seconds of the fluorescence signal recorded in the BCECF channel against that recorded in the Fura-2 or the RhoVR channel (**Figure 3C**, optical filtering). The slope of a linear fit

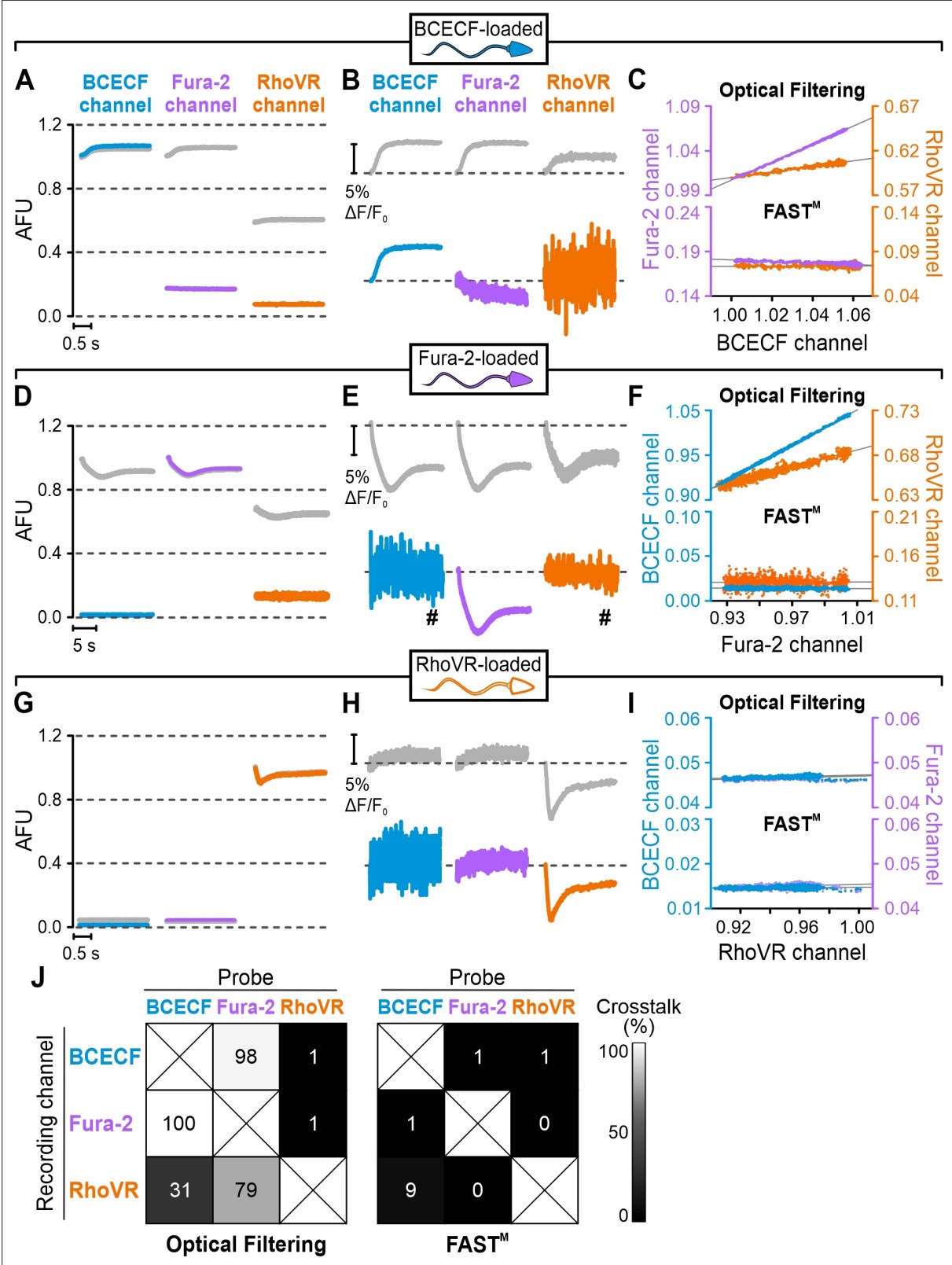

**Figure 3.** Resact-induced $pH_i$, $[Ca^{2+}]_i$, and $V_m$ signals recorded in sperm loaded with BCECF, Fura-2, or RhoVR using optical filtering alone or frequency- and spectrally-tuned multiplexing (FAST[M]). Time course of the fluorescence signals recorded from BCECF- (**A–C**), Fura-2- (**D–F**), or RhoVR-loaded sperm (**G–I**) after mixing with resact (50 pM). Fluorescence was recorded in the BCECF, Fura-2, and RhoVR channels using optical filtering alone (gray traces) or FAST[M] (colored traces). (**A, D, G**) Fluorescence signals in arbitrary fluorescence units (AFU); to ease the comparison, signals in (**A**), (**D**), and (**G**) were

*Figure 3 continued on next page*

*Figure 3 continued*

normalized (set to 1) to the baseline fluorescence ($F_0$) in the BCECF, the Fura-2, and the RhoVR channel, respectively, recorded immediately after mixing with resact. (**B, E, H**) Resact-evoked change in fluorescence ($\Delta F$) with respect to the baseline fluorescence ($F_0$), that is, $\Delta F/F_0$ (%); #signals smoothed with a sliding average of 80 ms. (**C, F, I**) First 2 s of the fluorescence signal recorded in the BCECF channel plotted against that recorded in the Fura-2 or the RhoVR channel using either optical filtering (top panel) or FAST$^M$ (bottom panel). Gray line: linear fit of the plots to quantify the crosstalk between the channels (see explanation in the text). (**J**) Percent crosstalk between the channels according to the analysis shown in (**C**), (**F**), and (**I**).

The online version of this article includes the following source data for figure 3:

**Source data 1.** Fluorescence signals in arbitrary fluorescence units.

to these plots is a measure of the crosstalk: if the time course of the fluorescence perfectly correlates between channels, the slope and crosstalk is 1 and 100%, respectively. Vice versa, if the time course of the fluorescence is independent among channels, the slope/crosstalk is zero. For optical filtering alone, we determined a crosstalk between the BCECF and the Fura-2 and RhoVR channels of 100 and 31%, respectively (*Figure 3J*). Modulating the LEDs at different frequencies using FAST$^M$ did not affect the fluorescence signal in the BCECF channel (*Figure 3A and B*, blue trace). However, FAST$^M$ lowered the basal fluorescence and almost abolished its relative increase in both the Fura-2 (*Figure 3A and B*; cyan) and the RhoVR channel (*Figure 3A and B*, orange); with FAST$^M$, the crosstalk between the BCECF and the Fura-2 or the RhoVR channel was only 9 and 1%, respectively (*Figure 3C and J*).

Next, we loaded sperm with Fura-2 alone and monitored the resact-induced $[Ca^{2+}]_i$ response. With optical filtering alone, the basal fluorescence and its relative decrease, reflecting the $[Ca^{2+}]_i$ response, were similar in all channels (*Figure 3D and E*; gray traces); the crosstalk between the Fura-2 and the BCECF or RhoVR channels was 98 and 79%, respectively (*Figure 3F and J*). Of note, Fura-2 fluorescence decreased with increasing $[Ca^{2+}]_i$ because the probe was excited at 380 nm. FAST$^M$ did not affect the Fura-2 channel (*Figure 3D and E*; cyan), but lowered the basal fluorescence intensity and abolished its relative decrease in the BCECF channel (*Figure 3D and E*; blue) and the RhoVR channel (*Figure 3D and E*; orange); the crosstalk between the channels was ≤1% (*Figure 3F and J*). Finally, we monitored the resact-induced $V_m$ response in RhoVR-loaded sperm. Due to the probe's red-shifted spectrum, crosstalk between channels was negligible; basal RhoVR fluorescence and its resact-induced decrease, reflecting the $V_m$ response, were only detected in the RhoVR channel, both with and without FAST$^M$ (*Figure 3G–J*).

We next loaded sperm with all three probes and simultaneously recorded resact-induced $[Ca^{2+}]_i$, $pH_i$, and $V_m$ signals (*Figure 4A and B*). Using optical filtering alone, the simultaneously recorded signals markedly differed from the respective signals recorded in sperm loaded with one probe only (compare *Figure 4A* and *Figure 3B, E,H* ); the $pH_i$ and $[Ca^{2+}]_i$ signals represent a composite of the Fura-2-reported $Ca^{2+}$ response (transient fluorescence decrease) and the BCECF-reported $pH_i$ response (sustained fluorescence increase), whereas the $V_m$ signal featured a lower amplitude and

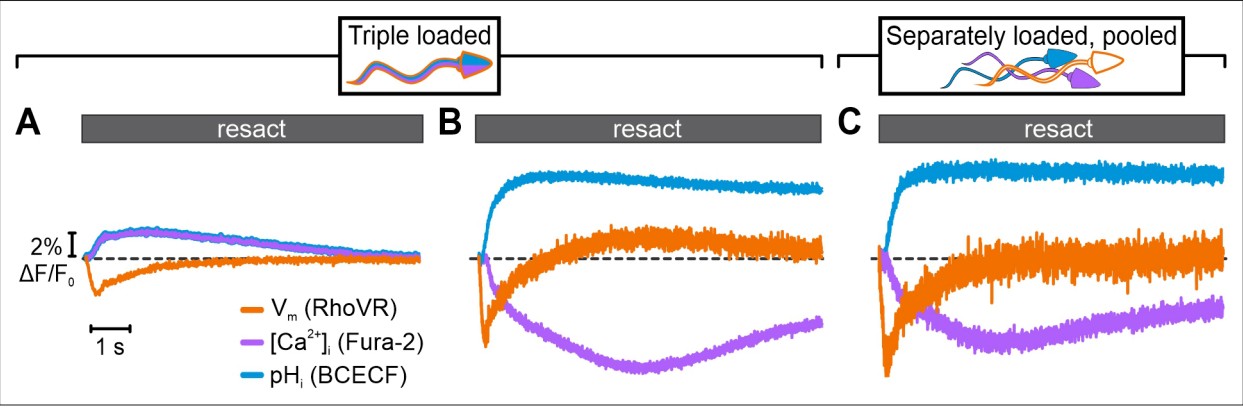

**Figure 4.** Simultaneous recording of resact-evoked $pH_i$, $[Ca^{2+}]_i$, and $V_m$ signals in sperm loaded with BCECF, Fura-2, and RhoVR. Relative changes in fluorescence $\Delta F/F_0$ evoked by 50 pM resact. The respective control signal evoked by mixing with artificial sea water (ASW) was subtracted, setting the control-signal level to $\Delta F/F_0$ (%) = 0 (dotted line). Signals were recorded using optical filtering alone (**A**) or frequency- and spectrally-tuned multiplexing (FAST$^M$) (**B**). (**C**) Simultaneous FAST$^M$ recording of resact-evoked signals from pooled sperm loaded separately with either BCECF, Fura-2, or RhoVR.

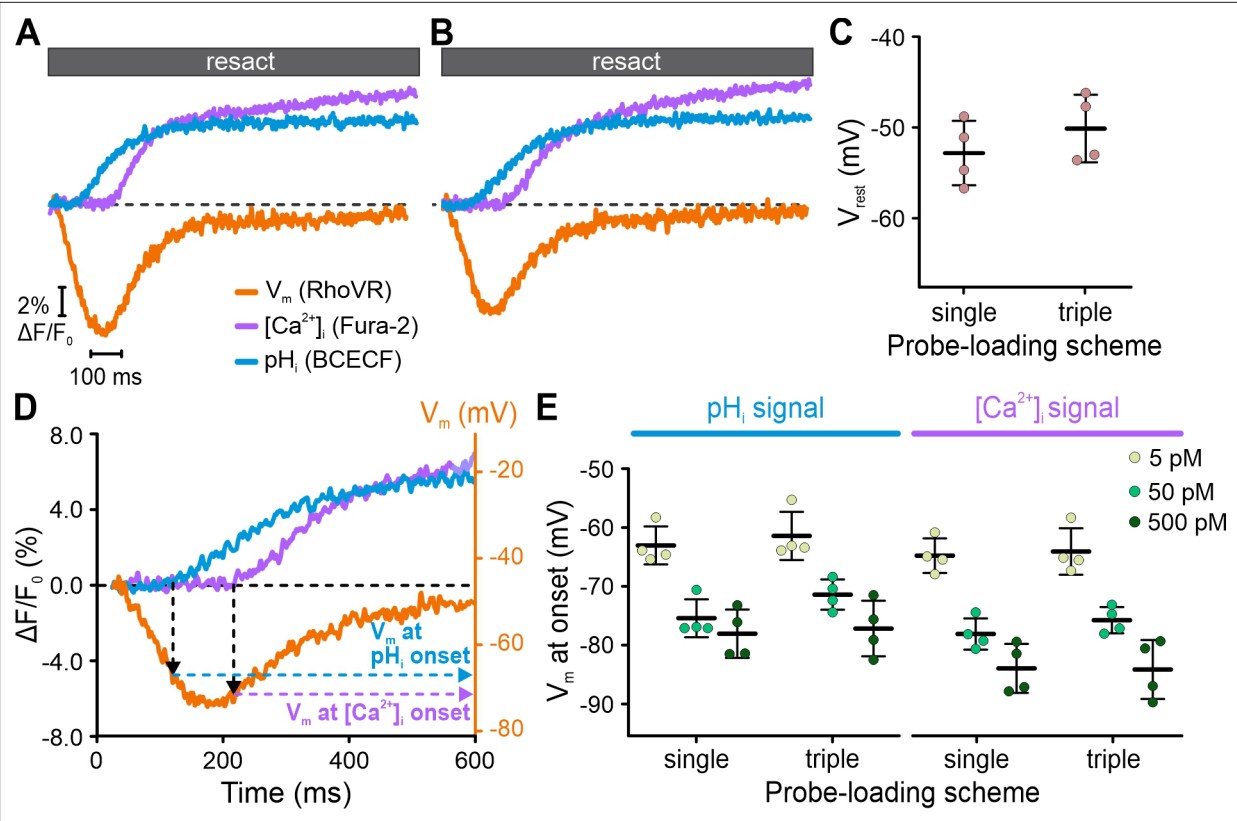

**Figure 5.** Interrogating putative probe-related perturbations of signaling. Resact-evoked $V_m$, $pH_i$, and $[Ca^{2+}]_i$ signals recorded individually from different sperm samples loaded with one probe only (**A**) or recorded simultaneously from triple-loaded sperm (**B**); to facilitate direct comparison, Fura-2 fluorescence was multiplied by –1 to depict an increase of $[Ca^{2+}]_i$ as an increasing signal. (**C**) Comparison of $V_{rest}$ of sperm loaded with RhoVR (single-loaded) or RhoVR, BCECF, and Fura-2 (triple-loaded). (**D**) Calibrated resact-induced (50 pM) $V_m$ response and accompanying $pH_i$ and $[Ca^{2+}]_i$ signals. The artificial sea water (ASW) control was subtracted, and the dotted black line indicates $\Delta F/F_0 = 0$ and $V_{rest}$. The $V_m$ at the onset of the $pH_i$ and $[Ca^{2+}]_i$ signals was deduced from the signal latencies. (**E**) $V_m$ at the onset of $pH_i$ and $[Ca^{2+}]_i$ signals in single- versus triple-loaded sperm. With increasing resact concentrations, the rise in $pH_i$ and $[Ca^{2+}]_i$ commenced at increasingly negative $V_m$ (*Seifert et al., 2015*).

slower kinetics (*Figure 4A*). Thus, the crosstalk among channels greatly misrepresented the true time course and size of signaling events.

By contrast, using FAST$^M$, we simultaneously recorded genuine resact-induced $[Ca^{2+}]_i$, $pH_i$, and $V_m$ signals in the respective channels (*Figure 4B*). The kinetics, waveforms, and amplitudes of the multiplexed signals were similar to those recorded with FAST$^M$ (compare *Figure 4B* with *Figure 3B, E and H*) or without FAST$^M$ (see previous studies, e.g., *Hamzeh et al., 2019*) in sperm loaded with one probe only. We further explored whether triple-loading per se affects the response waveforms. To this end, we pooled sperm suspensions that were separately loaded with either Fura-2, BCECF, or RhoVR. The overall time course of the $[Ca^{2+}]_i$, $pH_i$, and $V_m$ signals recorded simultaneously via FAST$^M$ from these pooled single-loaded sperm was similar to those recorded from triple-loaded sperm (*Figure 4C*). Competition of probes with downstream targets for signaling molecules might perturb response dynamics (*Lew et al., 1985*; *Haugh, 2012*; *Delvendahl et al., 2015*); in triple-loaded cells, this potential caveat might be enhanced. Therefore, using FAST$^M$, we further examined in greater detail whether specific features of the signals were altered in single- vs. triple-loaded sperm. We compared resact-induced $[Ca^{2+}]_i$, $pH_i$, and $V_m$ signals in sperm loaded with one probe (single-loaded) to those in sperm loaded with three probes (triple-loaded); of note, for the ease of illustration, Fura-2 fluorescence was multiplied by –1 to depict an increase of $[Ca^{2+}]_i$ as an increasing signal. Under both conditions, the respective signals were similar (*Figure 5A and B*). We took this comparison one step further and compared the resting membrane potential ($V_{rest}$) and threshold voltage ($V_{thr}$) at which $[Ca^{2+}]_i$ and $pH_i$ commence to rise after stimulation with different resact concentrations; $V_{rest}$ and $V_{thr}$ are characteristic features of the signaling pathway (*Figure 5C–E*; *Seifert et al., 2015*). In single- and

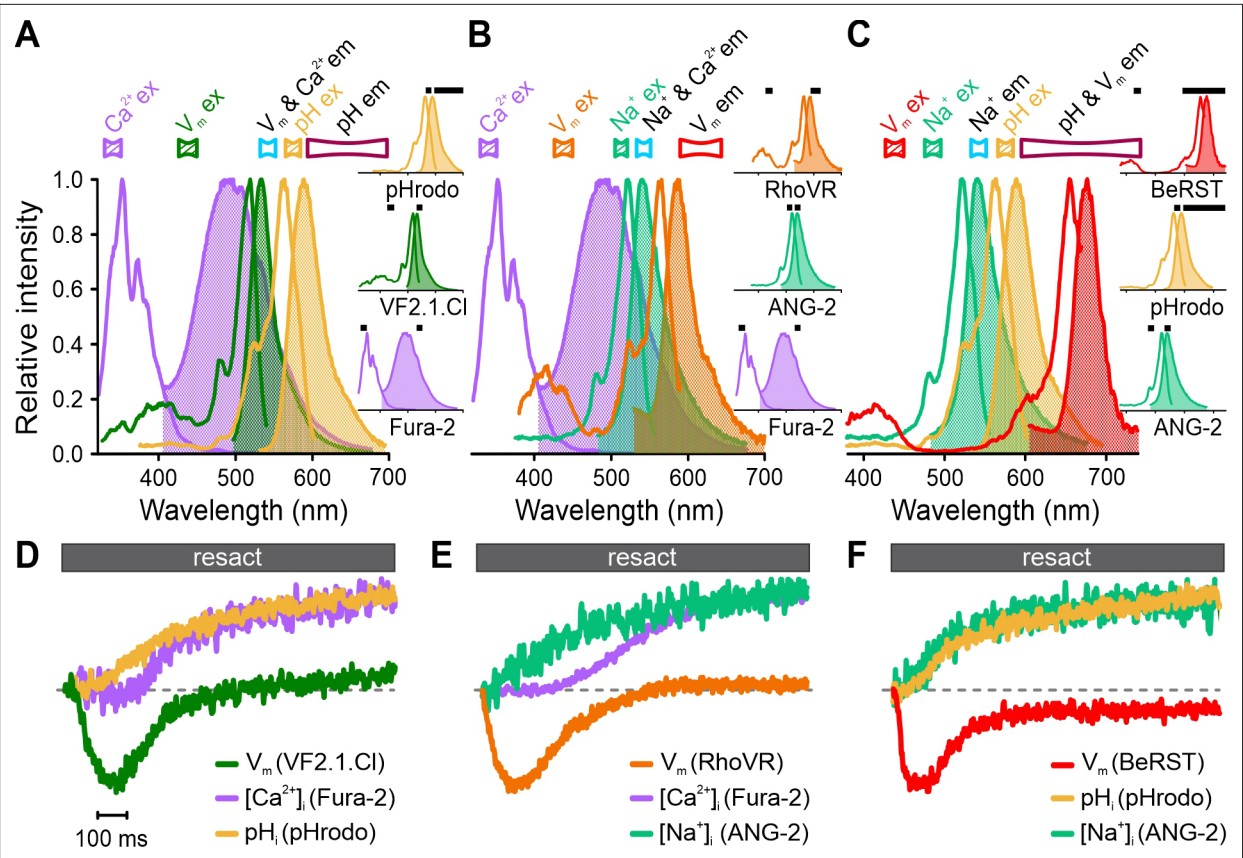

**Figure 6.** Simultaneous recording of resact-induced signaling events in sperm loaded with various triple combinations of probes for $Ca^{2+}$, pH, $Na^+$, and $V_m$ using frequency- and spectrally-tuned multiplexing (FAST$^M$). Superposition of excitation (outlined) and emission (filled) spectra of (**A**) Fura-2, VF2.1.Cl, and pHrodo; (**B**) Fura-2, RhoVR, and ANG-2; (**C**) BeRST, pHrodo, and ANG-2. Bandpass filters used for excitation (filled) and emission (outlined) are depicted above the spectra. Inset: individual excitation and emission spectra with respective filters (black bars). (**D–F**) Signals (ΔF/F$_0$) evoked by 500 pM resact corrected for the artificial sea water (ASW) control and normalized to their respective peak values (set to 1) for easier illustration.

The online version of this article includes the following figure supplement(s) for figure 6:

**Figure supplement 1.** Spectral utility of voltage-sensitive probes.

triple-loaded sperm, both $V_{rest}$ (*Figure 5C*) and $V_{thr}$ (*Figure 5D and E*) were similar. Thus, signaling is neither perturbed by $Ca^{2+}$- and $H^+$-binding to Fura-2 and BCECF, respectively, nor by partition of RhoVR into the membrane, at least under the experimental regimes used here.

We conclude that Fura-2, BCECF, and RhoVR are not suitable for simultaneous recording based on optical filtering alone, whereas FAST$^M$ permits this probe combination for multiplexing of rapid [$Ca^{2+}$]$_i$, pH$_i$, and $V_m$ responses with millisecond time resolution.

To illustrate the versatility of FAST$^M$, we tested different triple combinations of $V_m$, $Ca^{2+}$, pH, and $Na^+$ probes, whose overlapping emission spectra prevent simultaneous recording using optical filtering alone (*Figure 6A–C*, *Figure 6—figure supplement 1*). By contrast, FAST$^M$ allowed for crosstalk-free multiplexing of resact-induced $V_m$-[$Ca^{2+}$]$_i$-pH$_i$ (*Figure 6D*), $V_m$-[$Ca^{2+}$]$_i$-[$Na^+$]$_i$ (*Figure 6E*), or $V_m$-pH$_i$-[$Na^+$]$_i$ (*Figure 6F*) responses.

Finally, the shift of the excitation spectra of Fura probes and BCECF upon $Ca^{2+}$ and $H^+$ binding, respectively, can be harnessed to quantify [$Ca^{2+}$]$_i$ or pH$_i$ in absolute terms using ratiometric recording (*O'Connor and Silver, 2013*). This relies on obtaining the ratio of the probe's emission recorded at two different excitation wavelengths, which, in previous studies, required switching between excitation wavelengths. We investigated whether FAST$^M$ allows for simultaneous ratiometric recording of Fura-FF and BCECF. Moreover, we used human instead of sea urchin sperm, thus, testing FAST$^M$ in different cells. In human sperm, the CatSper $Ca^{2+}$ channel is activated at alkaline pH$_i$ and also by the female steroid hormone progesterone (*Lishko et al., 2011*; *Strünker et al., 2011*). We mixed

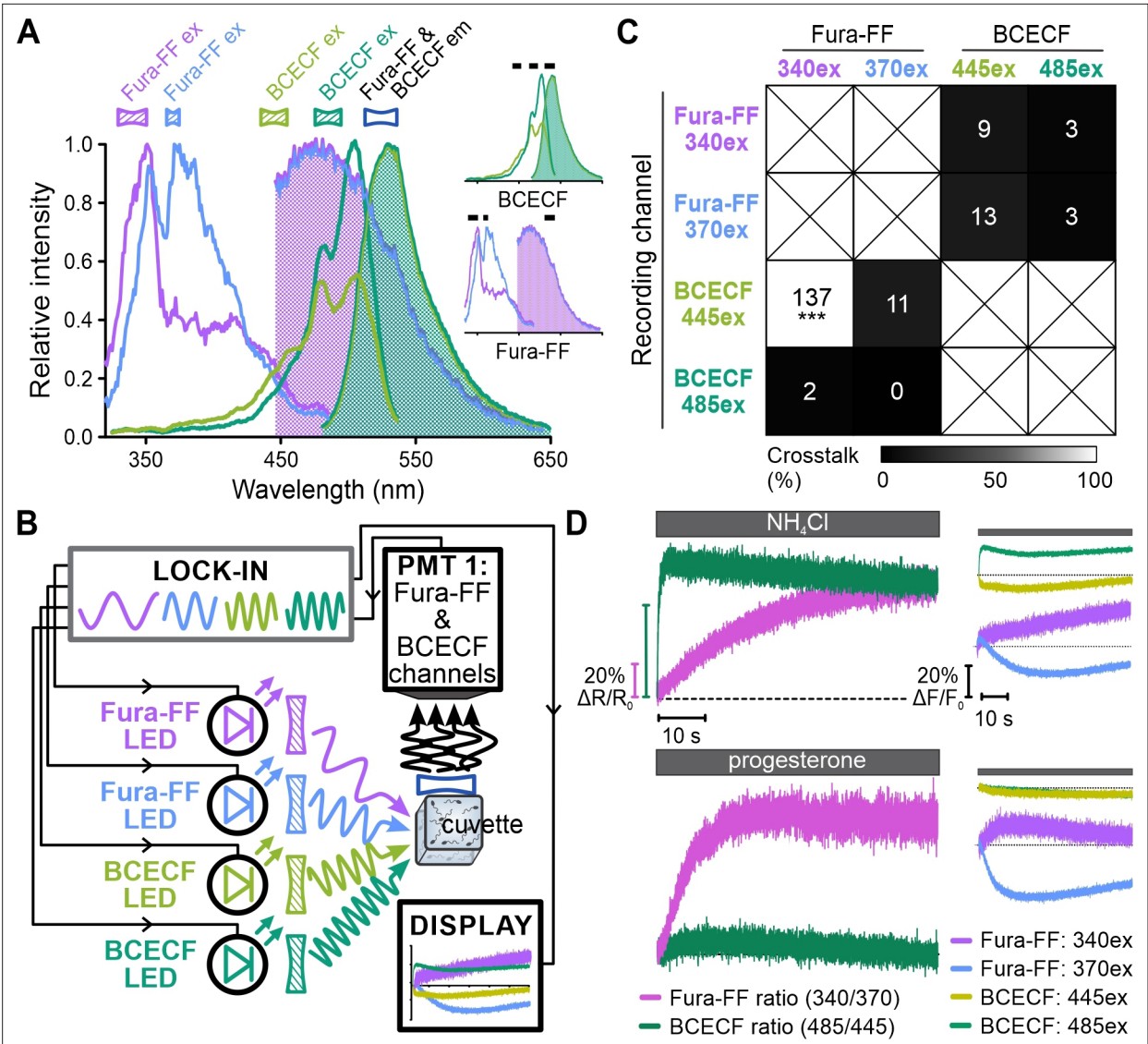

**Figure 7.** Simultaneous ratiometric recording of $[Ca^{2+}]_i$ and $pH_i$ signals in human sperm. (**A**) Superimposed excitation (outlined) and emission (filled) spectra of Fura-FF and BCECF. Inset: individual spectra with respective filters (black bars). (**B**) Schematic of frequency- and spectrally-tuned multiplexing (FAST^M) configuration for simultaneous ratiometric dual-excitation recording of Fura-FF and BCECF in human sperm. (**C**) Crosstalk among channels based on the analysis shown in *Figure 7—figure supplement 1*. #Under these particular conditions, the approach to quantify crosstalk yielded an erroneously inflated value (for details, see *Figure 7—figure supplement 1*). (**D**) Left panels: ratiometric $[Ca^{2+}]_i$ and $pH_i$ signals ($\Delta R/R_0$) in human sperm evoked by $NH_4Cl$ (10 mM) or progesterone (100 nM) corrected for the buffer control. Right panel: fluorescence signals in the individual Fura-FF and BCECF channels underlying the ratio.

The online version of this article includes the following source data and figure supplement(s) for figure 7:

**Figure supplement 1.** Analysis of crosstalk between BCECF and Fura-FF channels recorded with frequency- and spectrally-tuned multiplexing (FAST^M).

**Figure supplement 1—source data 1.** Fluorescence signals for the analysis of crosstalk between BCECF and Fura-FF channels.

Fura-FF- and BCECF-loaded human sperm in the stopped-flow device with $NH_4Cl$ or progesterone. Fura-FF and BCECF were simultaneously excited each at two different wavelengths (340/370 nm and 445/485 nm, respectively) with frequency-modulated light; the emission was collected at 530 nm by one detector (*Figure 7A and B*, Table 3). $NH_4Cl$ evoked an instantaneous, rapid and more gradual increase in the emission ratios of BCECF and Fura-2, respectively, reflecting the $NH_4Cl$-induced $pH_i$ increase and concomitant $pH_i$-induced $Ca^{2+}$ influx via CatSper, respectively (*Figure 7D*). By contrast, progesterone evoked an instantaneous increase of the Fura-FF ratio, reflecting progesterone-induced $Ca^{2+}$ influx, whereas the BCECF ratio was largely unaffected (*Figure 7D*). These results demonstrate

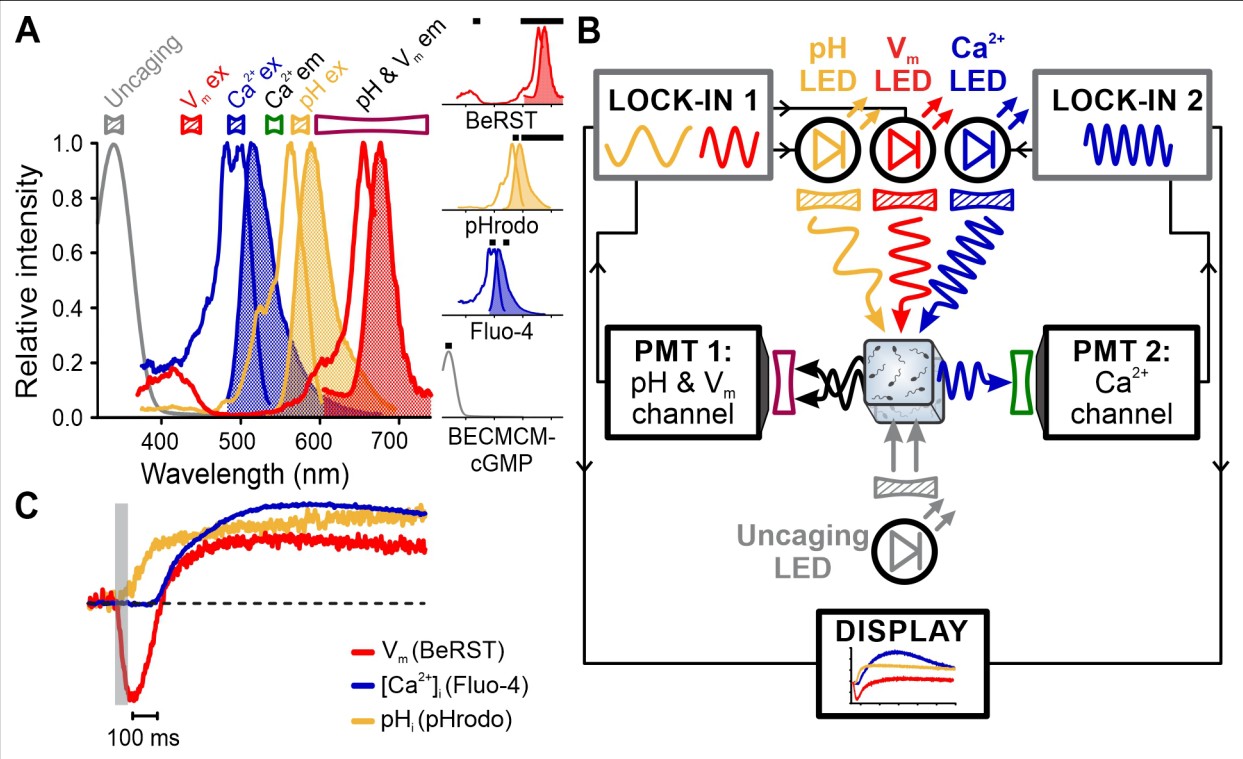

**Figure 8.** Simultaneous recording of $[Ca^{2+}]_i$, $pH_i$, and $V_m$ signals in sea urchin sperm evoked by flash photolysis of caged cGMP. (**A**) Superimposed absorbance spectrum of BECMCM-cGMP and excitation (outlined) and emission (filled) spectra of Fluo-4, pHrodo, and BeRST. Bandpass filters used for excitation (filled) and emission (outlined) are depicted above the spectra. Inset: individual spectra and respective filters (black bars). (**B**) Schematic of frequency- and spectrally-tuned multiplexing (FAST$^M$) configuration for uncaging experiments. (**C**) $V_m$, $pH_i$, and $[Ca^{2+}]_i$ signals evoked by uncaging intracellular cGMP with a 50 ms UV-flash (gray bar).

that FAST$^M$ ensures minimal crosstalk between the Fura-FF and the BCECF channels (**Figure 7C**, **Figure 7—figure supplement 1**). Taken together, FAST$^M$ allows for simultaneous recording of rapid signaling events with millisecond temporal resolution using various combinations of non-ratiometric and ratiometric probes.

## Combination of FAST$^M$ with flash photolysis of caged compounds

Optogenetics and optochemistry employ light-triggered tools (e.g., enzymes, ion channels, caged compounds, photoswitches) to investigate cellular signaling pathways (**Ellis-Davies, 2007**; **Szymański et al., 2013**, **Ankenbruck et al., 2018**). In general, combining such tools with fluorescent probes requires shielding the detectors from the trigger such as the strong UV flash used for uncaging. Optical filtering alone is usually not sufficient to prevent recording artifacts created by the UV flash (e.g., see **Strünker et al., 2006**; **Kilic et al., 2009**; **Servin-Vences et al., 2012**). We used sea urchin sperm to explore whether FAST$^M$ can ameliorate flash artifacts. Sperm were loaded with Fluo-4, pHrodo, BeRST, and BECMCM-caged cGMP to simultaneously record $[Ca^{2+}]_i$, $pH_i$, and $V_m$ responses evoked by the intracellular photorelease of cGMP that bypasses receptor GC activation (**Hamzeh et al., 2019**; **Figure 8A and B**). Indeed, due to the phase-sensitive signal detection, the flash artifact was suppressed by the lock-in amplifiers (**Hamzeh et al., 2019**) and FAST$^M$ allowed for undisturbed simultaneous recording of cGMP-evoked $[Ca^{2+}]_i$, $pH_i$, and $V_m$ signals (**Figure 8C**).

## Multiplexing of fast chemical reactions using FAST$^M$

We next explored whether FAST$^M$ also allows multiplexing of fast chemical reactions in solution. Using the stopped-flow device, we simultaneously monitored the kinetics of $Ca^{2+}$ dissociation from Fura-2 (dual-excitation recording), Fluo-4, and Calbryte 630 (**Figure 9A–C**, **Figure 9—figure supplement 1**). A solution containing $Ca^{2+}$-bound Fura-2, Fluo-4, and Calbryte 630 was mixed with an excess of the $Ca^{2+}$ chelator BAPTA that competes with the probes for binding of $Ca^{2+}$. Dissociation of $Ca^{2+}$ was

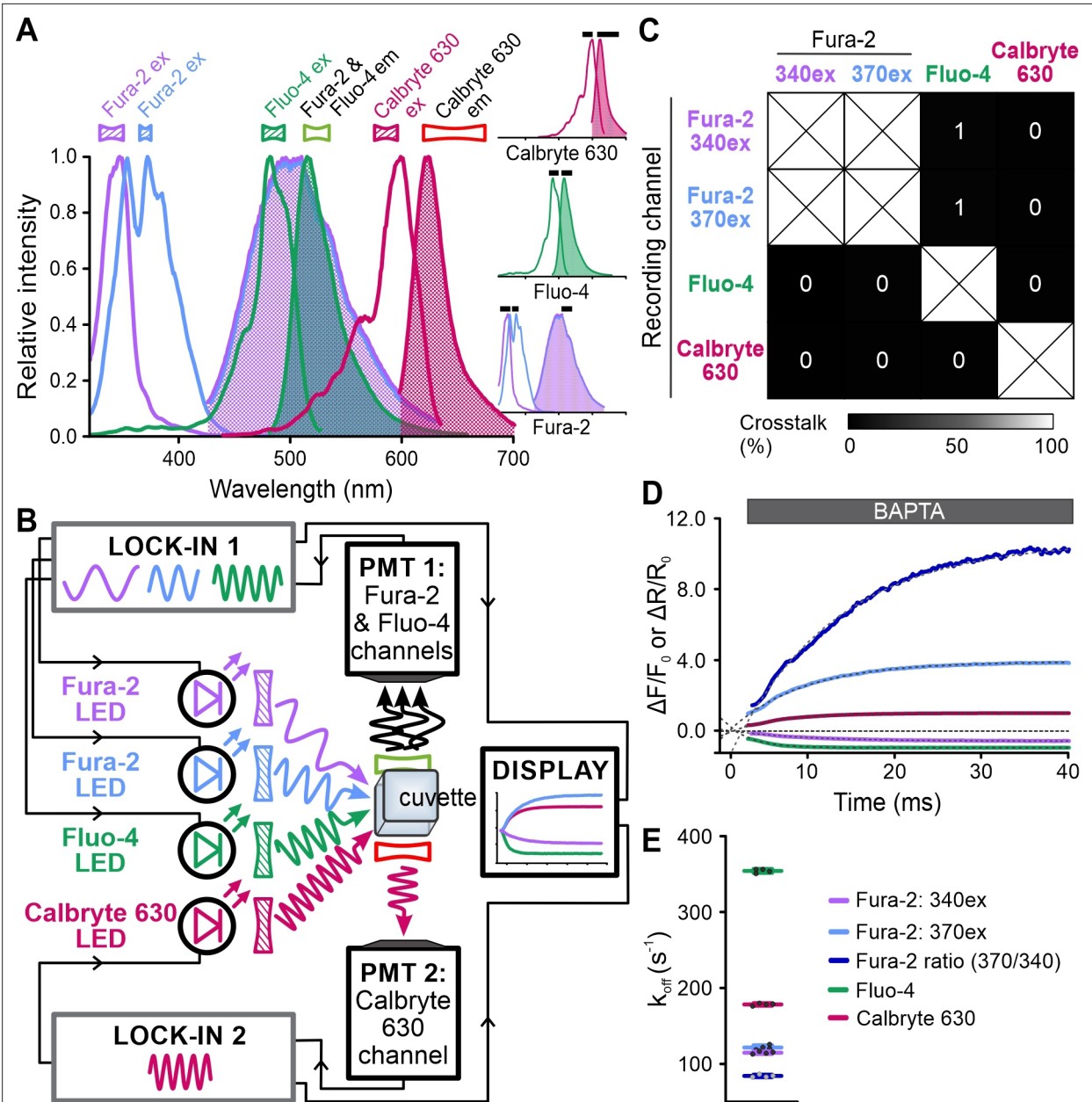

**Figure 9.** Simultaneous recording of the kinetics of $Ca^{2+}$ dissociation from Fura-2, Fluo-4, and Calbryte 630. (**A**) Superimposed excitation (outlined) and emission (filled) spectra of Fura-2, Fluo-4, and Calbryte 630. Inset: individual spectra depicted with respective filters (black bars). (**B**) Schematic of the frequency- and spectrally-tuned multiplexing (FAST[M]) configuration for simultaneous recording of $Ca^{2+}$ dissociation from Fura-2, Fluo-4, and Calbryte 630. (**C**) Crosstalk between channels according to *Figure 9—figure supplement 1*. (**D**) Changes in Fura-2, Fluo-4, and Calbryte 630 fluorescence and 370 nm/340 nm emission ratio of Fura-2 upon mixing of the $Ca^{2+}$-bound probes with the $Ca^{2+}$ chelator BAPTA. (**E**) $K_{off}$ values determined by exponential fitting of the individual fluorescence traces and the ratio of Fura-2 (370/340 nm): Fura-2, 340ex: $115 \pm 2$ $s^{-1}$; Fura-2, 370ex: $122 \pm 3$ $s^{-1}$; Fura-2 ratio: $84 \pm 2$ $s^{-1}$; Fluo-4: $354 \pm 3$ $s^{-1}$; Calbryte 630: $178 \pm 2$ $s^{-1}$ (n = 4).

The online version of this article includes the following source data and figure supplement(s) for figure 9:

**Figure supplement 1.** Analysis of crosstalk between Fura-2, Fluo-4, and Calbryte 630 channels simultaneously recorded with FAST[M].

**Figure supplement 1—source data 1.** Fluorescence signals for the analysis of crosstalk between Fura-2, Fluo-4, and Calbryte 630 channels.

**Figure supplement 2.** Signal-to-noise (S/N) ratio of fluorescence signals recorded upon mixing of $Ca^{2+}$-bound Fura-2 with BAPTA.

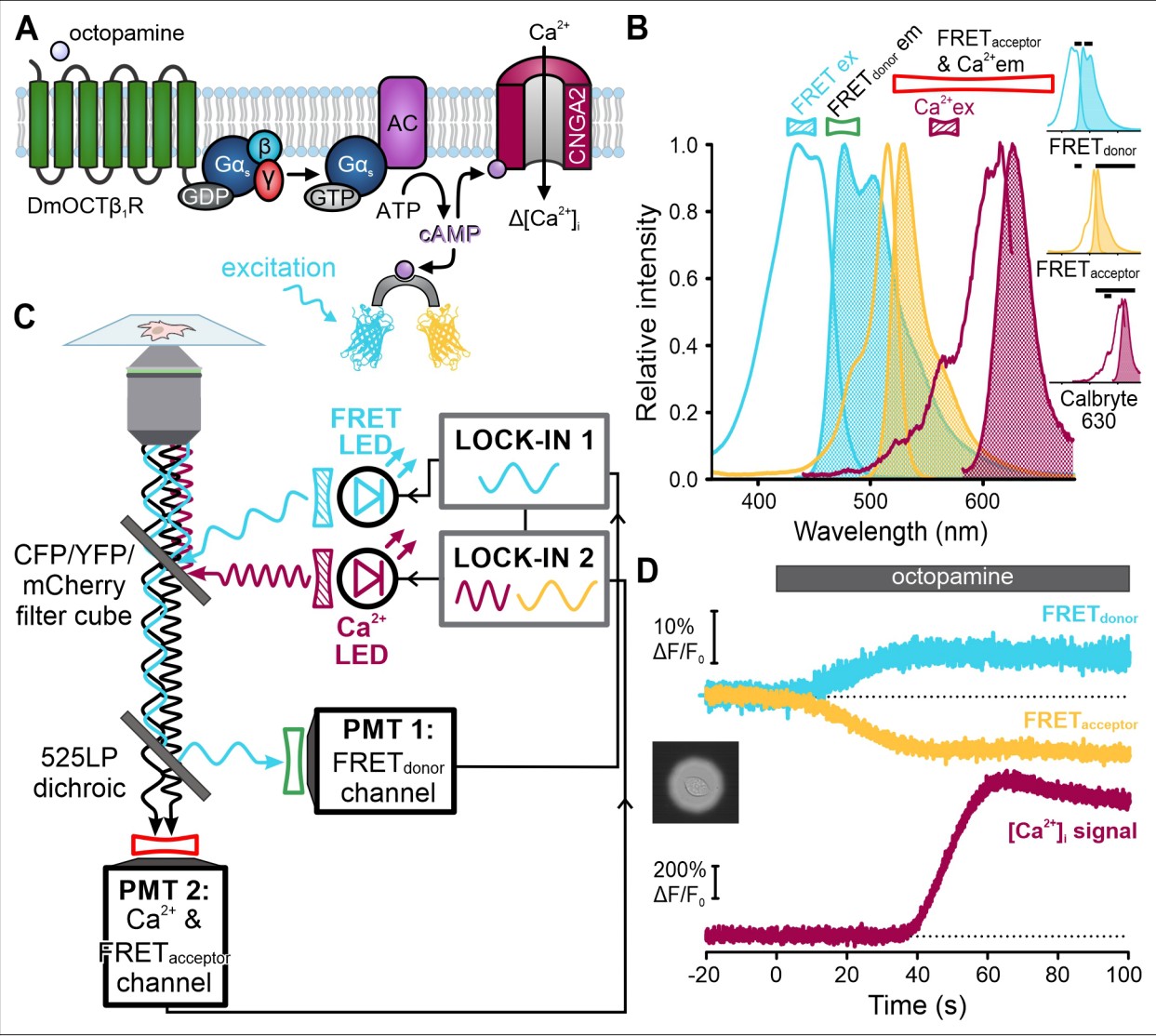

**Figure 10.** Single-cell frequency- and spectrally-tuned multiplexing (FAST$^M$) fluorescence microscopy for simultaneous recording of cAMP and $[Ca^{2+}]_i$. (**A**) Octopamine-signaling pathway in HEK cells coexpressing the DmOCTβ1 receptor, CNGA2-TM channel, and a FRET-based cAMP biosensor. (**B**) Superimposed excitation (outlined) and emission (filled) spectra of the FRET donor-acceptor pair cerulean-citrine, and the $Ca^{2+}$-probe Calbryte 630. Bandpass filters used for excitation (filled) and emission (outlined) are depicted above the spectra. Inset: individual spectra and filters (black bars). (**C**) Schematic of the FAST$^M$ configuration for single-cell microscopy. (**D**) Changes in fluorescence ($\Delta F/F_0$ (%)) of the FRET donor and acceptor as well as Calbryte 630 evoked by octopamine (20 μM). Inset: image of the field of view with a single cell enclosed by an aperture.

reflected by a decrease of Fluo-4 and Calbryte 630 fluorescence; Fura-2 fluorescence decreased and increased at 370 nm and 340 nm excitation, respectively (**Figure 9D**). Exponential fitting of the traces yielded the dissociation rate constants ($k_{off}$) (**Figure 9E**). The $k_{off}$ of Fura-2 ($340_{ex}$: 115 ± 2; $370_{ex}$: 122 ± 3; ratio: 84 ± 2 s$^{-1}$) was similar to that reported before (**Jackson et al., 1987**; **Kao and Tsien, 1988**), whereas that of Fluo-4 (354 ± 3 s$^{-1}$) and Calbryte 630 (178 ± 2 s$^{-1}$) (n = 4) had not yet been determined to the best of our knowledge. These experiments demonstrate the utility of FAST$^M$ for multiplexing of rapid chemical reactions.

## Single-cell FAST$^M$ fluorescence microscopy

Finally, we tested FAST$^M$ for single-cell fluorescence microscopy (**Figure 10**). A G$_s$-coupled octopamine receptor (DmOCTβ1R) (**Balfanz et al., 2005**), a FRET-based cAMP biosensor (**Mukherjee et al., 2016**), and a $Ca^{2+}$-permeable cyclic nucleotide-gated channel (CNGA2-TM) (**Wachten et al., 2006**; **Schröder-Lang et al., 2007**) were expressed in HEK293 cells (**Figure 10A**). Changes in octopamine-induced

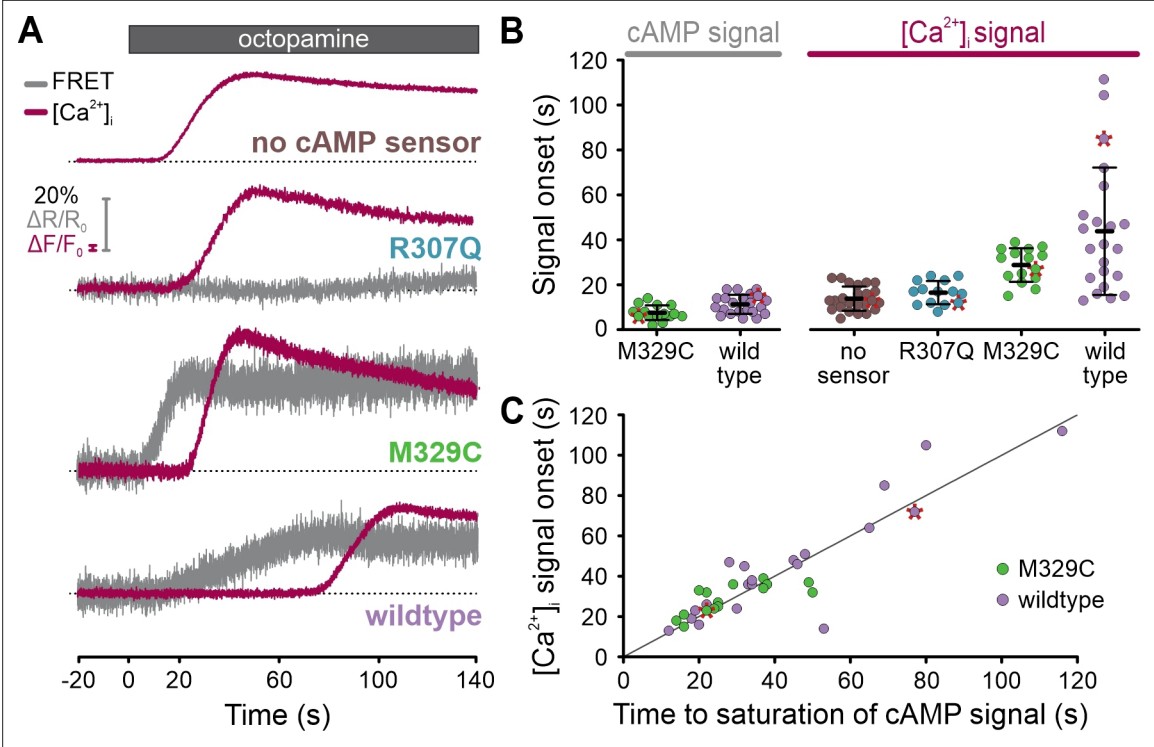

**Figure 11.** Simultaneous recording of cAMP and $[Ca^{2+}]_i$ signals in single cells using frequency- and spectrally-tuned multiplexing (FAST$^M$). (**A**) Octopamine-induced (20 µM) $[Ca^{2+}]_i$ signals and changes in the FRET ratio (donor/acceptor), that is, cAMP signals, in the absence or presence of a non-binding (R307Q), lower (M329C), or higher-affinity (wildtype) FRET-based cAMP biosensor. Data points corresponding to the representative traces are labeled with a red asterisk in (**B**) and (**C**). (**B**) Onset of the octopamine-induced cAMP and $[Ca^{2+}]_i$ signals. (**C**) Comparison of the time to saturation of the cAMP signal and the onset of the $[Ca^{2+}]_i$ signal. The gray line depicts the theoretical perfect correlation.

The online version of this article includes the following source data and figure supplement(s) for figure 11:

**Source data 1.** Onset of the optopamine-induced cAMP and $[Ca^{2+}]_i$ signals.

**Figure supplement 1.** Affinity of mlCNBD-M329C for 8-NBD-cAMP.

cAMP synthesis and cAMP-induced $Ca^{2+}$ influx were simultaneously recorded using the cAMP biosensor and the $Ca^{2+}$ probe Calbryte 630, respectively (***Figure 10B***). The FRET donor (cerulean) and Calbryte 630 were excited by light modulated at different frequencies (***Figure 10C***). Because the emission from the FRET donor and acceptor (citrine) was encoded with the same frequency, signal discrimination was achieved by optically filtering the cerulean and citrine fluorescence and collecting with separate detectors. Calbryte 630 and citrine fluorescence was collected by the same detector and discriminated by the modulation frequencies. Octopamine increased and decreased the donor and acceptor fluorescence of the FRET sensor, respectively, indicating a rise of intracellular cAMP (***Figure 10D***). Subsequently, Calbryte 630 fluorescence increased, indicating cAMP-induced $Ca^{2+}$ influx (***Figure 10D***). These results demonstrate that FAST$^M$ can also be employed for multiplexing in single cells using fluorescence microscopy and protein-based FRET sensors.

Curiously, $[Ca^{2+}]_i$ did not rise until the FRET signal reached saturation (***Figure 10D***), indicating that the cAMP sensor competes with the channel for cAMP. The vastly different $K_{1/2}$ values – about 70 nM (FRET sensor) vs. 10 µM (CNGA2-TM channel) – argue that the FRET sensors get served first, which might affect cAMP dynamics and thus CNGA2-TM activation. We tested this presumption and measured the latency of the $[Ca^{2+}]_i$ signal in cells lacking or expressing different variants of the cAMP sensor. The latency was similar in cells lacking the cAMP sensor (14 ± 5 s; n = 24) or expressing a sensor mutant (***Mukherjee et al., 2016***) that does not bind cAMP (mlCNBD-FRET-R307Q, 16 ± 5 s; n = 14) (***Figure 11A and B***). By contrast, the latency increased considerably and coincided with the saturation of the cAMP signal in cells expressing either the high-affinity cAMP sensor (44 ± 28 s, n = 22; mlCNBD-FRET) or a variant with lower cAMP affinity (~1 µM) (29 ± 7 s, n = 15; mlCNBD-FRET-M329C) (***Figure 11A-C***; ***Figure 11—figure supplement 1***). These findings support the notion that

cAMP 'buffering' by the cAMP sensor delayed activation of the downstream effector CNGA2-TM. This application of FAST[M] in single cells illustrates how probe-related perturbations of signaling pathways can be unveiled by multiplexing experiments.

## Discussion

We show that phase-sensitive signal detection using FAST[M] readily overcomes the fluorescence cross-talk that has limited true simultaneous recording of probes. The technical implementation of FAST[M] is straightforward: most LED-based light sources can be modulated in the kHz range, and conventional PMT-based fluorescence-detection setups can readily be upgraded with a lock-in amplifier featuring several demodulation channels. FAST[M] could be further advanced using additional LEDs, modulation frequencies, and detectors as well as optimized optical filtering. This will allow simultaneous recording of even more than three probes and four fluorescence 'channels' (e.g., for dual-excitation recording of Fura-FF and BCECF) as used here. Importantly, the temporal resolution of FAST[M] is largely independent of the number of probes and only limited by the time constant of the lock-in amplifier(s) and/ or detector(s), allowing multiplexing with a time resolution of a few microseconds. For comparison, using state-of-the-art filter wheels and galvanometer-based devices, quasi-simultaneous recording of four 'channels' can be performed with a time resolution of only >150 ms and >20 ms, respectively, which would be insufficient to fully resolve such rapid signaling events and chemical reactions that we studied here. Furthermore, phase-sensitive signal detection increases the signal-to-noise (S/N) ratio (e.g., *Meade, 1983*; *Figure 9—figure supplement 2*); thus, using FAST[M], reasonable S/N ratios can be reached with lower light intensity and density of fluorophores, which minimizes bleaching and sample consumption. Therefore, we envisage that FAST[M] will be widely adopted for simultaneous recording of rapid signals in aqueous solutions, single cells, and cell populations.

In combination with stopped-flow techniques and optochemical tools, the exceptional time resolution of FAST[M] might not only allow simultaneous recording of rapid chemical reactions, but also ligand-binding kinetics and the ensuing conformational changes of a protein (e.g., *Cukkemane et al., 2007*; *Peuker et al., 2013*). Cognate ligands labeled with solvatochromic fluorophores that change their fluorescence upon binding could be rapidly mixed with proteins. Ligand binding and the protein's conformational change could be simultaneously recorded by means of fluorescence or absorbance from endogenous tryptophan residues, incorporated non-natural amino acids, extrinsic fluorescent labels, or combinations thereof (*Cheng et al., 2020*).

Protein conformations and protein-protein interactions in macromolecular complexes are often investigated by time-resolved FRET (trFRET) (*Miyawaki and Niino, 2015*). Time-resolved readout of several FRET pairs is challenging as it requires two spectrally separated fluorophores for each FRET pair (*Depry et al., 2013*). Crosstalk arising from overlapping emission spectra compromises discrimination of FRET pairs; therefore, multiplexed trFRET measurements are susceptible to artifacts. The simultaneous recording of ratiometric probes using FAST[M] provides ample opportunities for future trFRET measurements with millisecond temporal resolution.

Finally, we show the applicability of FAST[M] for simultaneous recording of probes in single cells. However, the FAST[M] approach lacked spatial information. Subcellular imaging with FAST[M] could be achieved using PMT arrays. Alternatively, fluorescence could be recorded from several regions of interest (ROIs), whereby each ROI is illuminated with light modulated at a distinct frequency using acousto-optic-modulated laser excitation (*Wu et al., 2006*) or rapid switching of a digital micromirror device (DMD) (*Chang et al., 2016*; *Wang et al., 2016*). This approach would allow recording fluorescence signals from different ROIs with one PMT. Fast modulation of the excitation laser combined with a fast lock-in amplifier renders even confocal microscopes compatible with FAST[M] (*Carlsson et al., 1994*; *Wu et al., 2006*).

## Materials and methods

**Key resources table**

| Reagent type (species) or resource | Designation | Source or reference | Identifiers | Additional information |
|---|---|---|---|---|
| Cell line (HEK293) | flp-In-293 | Invitrogen | #R750-07 | RRID:CVCL_U421 |

*Continued on next page*

| Reagent type (species) or resource | Designation | Source or reference | Identifiers | Additional information |
|---|---|---|---|---|
| Transfected construct (*Drosophila melanogaster*) | DmOCTβ1R | *Balfanz et al., 2005* | | |
| Transfected construct (*Bos taurus*) | CNGA2-TM | *Schröder-Lang et al., 2007* | | |
| Recombinant DNA reagent | pc3.1-ml CNBD-FRET | *Mukherjee et al., 2016* | | |
| Recombinant DNA reagent | pc3.1-mlCNBD-FRET-R307Q | *Mukherjee et al., 2016* | | |
| Recombinant DNA reagent | pc3.1-mlCNBD-FRET-M329C | This paper | | *Figure 11—figure supplement 1* and materials and methods part of this MS. |
| Other | Pluronic F-127 | Sigma-Aldrich | P2443 | |
| Other | Fluo-4 AM | Thermo Fisher | F14202 | |
| Other | BCECF AM | Thermo Fisher | B1150 | |
| Other | Fura-2 AM | Thermo Fisher | F1201 | |
| Other | pHrodo Red AM | Thermo Fisher | P35372 | |
| Other | ANG-2 AM | MobiTec | 3502 | |
| Other | Calbryte 630 AM | AAT Bioquest | 20720 | |
| Other | Fura-FF, AM | AAT Bioquest | 21027 | |
| Other | VF2.1.Cl | *Miller et al., 2012* | Sold by Thermo Fisher as *FluoVolt* | |
| Other | BeRST | *Huang et al., 2015* | | |
| Other | RhoVR | *Deal et al., 2016* | | |
| Other | Calbryte 630, potassium salt | AAT Bioquest | 20727 | |
| Other | Fluo-4, pentapotassium salt | Thermo Fisher | F14200 | |
| Other | Fura-2, pentapotassium salt | Thermo Fisher | F1200 | |

## Handling of sperm from *Arbacia punctulata*

The protocol for sperm collection and the composition of artificial sea water (ASW) was described previously (*Hamzeh et al., 2019*). In short, spawning was induced by injection of 500 µl of 0.5 M KCl into the body cavity, and the spawn (dry sperm) was collected with a Pasteur pipette and stored on ice.

## Measurement of changes in $[Ca^{2+}]_i$, $pH_i$, $[Na^+]_i$, and $V_m$

Changes in $[Ca^{2+}]_i$, $pH_i$, $[Na^+]_i$, and $V_m$ in sea urchin sperm were measured in a rapid-mixing device (SFM-4000, FC-15 cuvette, BioLogic) in the stopped-flow mode. Dry sperm were loaded with fluorescent probe(s) according to *Table 1*. Probes were added individually or sequentially to dry sperm diluted 1:6 (v/v) in ASW supplemented with 0.05% Pluronic F-127 (Sigma-Aldrich). After incubation at 18°C in the dark, the probe-loaded sperm suspension was diluted 1:20 (v/v) with ASW and allowed to equilibrate for 5 min prior to measurement. In the stopped-flow, the probe-loaded sperm suspension was mixed 1:1 (v/v) with ASW or ASW supplemented with resact at a flow rate of 1, 2, or 4 ml/s, resulting in a dead time of 36.6, 18.3, or 9.1 ms, respectively. The lead time on the hard-stop valve was 2 ms. The concentration of resact is given as the final concentration after mixing. BECMCM-cGMP was synthesized by Andreas Rennhack (Research Centre casear); the VoltageFluor (VF) probes, VF2.1.Cl, BeRST, and RhoVR, were synthesized in the lab of Evan Miller at UC Berkeley, according to published protocols (*Miller et al., 2012*; *Huang et al., 2015*; *Deal et al., 2016*). VF probes are based

**Table 1.** Loading protocols for fluorescent probes in *A. punctulata* sperm and FAST^M modulation frequencies.

| | Fura-2, BCECF, RhoVR | | | ANG-2, pHrodo, BeRST | | |
|---|---|---|---|---|---|---|
| Loading order | First | Second | Third | First | Second | Third |
| Name | Fura-2 AM | RhoVR | BCECF AM | ANG-2 AM | pHrodo Red AM | BeRST |
| Probe type | Ca$^{2+}$ | V$_m$ | pH | Na$^+$ | pH | V$_m$ |
| Concentration (µM) | 10 | 5 | 5 | 10 | 10 | 5 |
| Incubation (min) | 90 | 10 | 5 | 90 | 25 | 10 |
| *FAST^M modulation frequency (kHz)* | 30.4 | 37.3 | 50 | 37 | 50 | 23 |
| | Fura-2, ANG-2, RhoVR | | | Fura-2, pHrodo, VF2.1.Cl | | |
| Loading order | First | Second | Third | First | Second | Third |
| Name | Fura-2 AM | ANG-2 AM | RhoVR | Fura-2 AM | pHrodo Red AM | VF2.1.Cl |
| Probe type | Ca$^{2+}$ | Na$^+$ | V$_m$ | Ca$^{2+}$ | pH | V$_m$ |
| Concentration (µM) | 10 | 10 | 5 | 10 | 10 | 5 |
| Incubation (min) | 90 (added together) | | 5 | 50 | 35 | 5 |
| *FAST^M modulation frequency (kHz)* | 50 | 23 | 37 | 50.3 | 25.3 | 17.1 |
| | Fluo-4, phrodo, BeRST, BECMCM-cGMP | | | | |
| Loading order | First | Second | Third | Fourth |
| Name | BECMCM-cGMP | Fluo-4 AM | pHrodo Red AM | BeRST |
| Probe type | Caged cGMP | Ca$^2$ | pH | V$_m$ |
| Concentration (µM) | 10 | 10 | 10 | 5 |
| Incubation (min) | 15 | 10 | 35 | 10 |
| *FAST^M modulation frequency (kHz)* | None | 37.3 | 30.1 | 50.3 |

on photo-induced electron transfer and exhibit a principal absorbance peak and a secondary peak at ~400 nm. In multiplexed configurations, either peak was effectively employed to excite the VF probe and monitor V$_m$ (*Figure 6—figure supplement 1*).

Fluorescence was excited by an array of LEDs (Thorlabs) fitted with dichroics (*Table 2*). Lock-in amplifiers (MFLI, Zurich Instruments, and SR844 RF, Stanford Research Systems) supplied signals to modulate the LEDs, which were operated by a custom-made LED driver. Modulation frequencies were between 10 and 50 kHz (*Table 1*). The modulated output of the LEDs was combined with appropriate dichroics (*Table 2*) into a liquid light guide (series 380, Ø 3 mm × 1000 mm, Lumatec) and delivered to the cuvette (FC-15, BioLogic).

The emission was collected at right angles to the cuvette and spectrally filtered with bandpass filters (*Table 2*) onto two PMT modules (H10723-20; Hamamatsu Photonics). Signals from the PMTs were directed to lock-in amplifiers, where they were amplified and frequency filtered with a third-order (18 dB/octave) lowpass filter and a time constant of 1 ms. Data acquisition was performed with a data acquisition pad (PCI-6221; National Instruments) and Bio-Kine software (BioLogic) with a sampling rate of 1 or 2 ms. Of note, to investigate signals recorded upon optical filtering alone (*Figure 3*), all LEDs were modulated with the same frequency.

## Analysis of stopped-flow recordings from *A. punctulata* sperm

Data were analyzed using GraphPad Prism 9 (Prism, La Jolla, USA). Each signal represents the average of at least three recordings. Signals are depicted as the percent change in fluorescence with respect to the mean of the first 5–10 data points prior to signal onset (ΔF/F$_0$ (%)); mixing artifacts occurring at the stop of the flow were cropped, prolonging the actual dead time of recordings. The baseline ΔF/F$_0$ obtained upon mixing with ASW alone was subtracted wherever indicated. In some figures, signals

**Table 2.** Optical configurations for recording signals from *A. punctulata* sperm.

| Probe combination | Fluorescent probe | LED (Thorlabs) | Excitation filter (Semrock) | Dichroics | Emission filter (Semrock) |
|---|---|---|---|---|---|
| Fura-2, BCECF, RhoVR | Fura-2 | M375L4 | 379/34 | 470 LPXR (Chroma) HC BS 409 (Semrock) | 524/24 |
| | BCECF | M490L4 | 485/20 | | |
| | RhoVR | M455L4 | 438/24 | | 607/36 |
| ANG-2, pHrodo, BeRST | ANG-2 | M490L4 | 485/20 | 525 LPXR (Chroma) 470 LPXR (Chroma) | 542/20 |
| | pHrodo | M565L3 | 575/19 | | |
| | BeRST | M455L4 | 438/24 | | 593LP |
| Fura-2, ANG-2, RhoVR | Fura-2 | M340L4 | 340/22 | 470 LPXR (Chroma) HC BS 409 (Semrock) | 542/20 |
| | ANG-2 | M505L3 | 513/17 | | |
| | RhoVR | M455L4 | 438/24 | | 607/36 |
| Fura-2, pHrodo, VF2.1.Cl | Fura-2 | M340L4 | 340/22 | 470 LPXR (Chroma) HC BS 409 (Semrock) | 542/20 |
| | VF2.1.Cl | M455L4 | 438/24 | | |
| | pHrodo | M565L3 | 575/19 | | 593LP |
| BECMCM-cGMP, Fluo-4, pHrodo, BeRST | BECMCM-cGMP | M340L4 | 340/22 | 525 LPXR (Chroma) 470 LPXR (Chroma) HC BS 409 (Semrock) | |
| | Fluo-4 | M490L4 | 494/20 | | 542/20 |
| | pHrodo | M565L3 | 575/19 | | |
| | BeRST | M455L4 | 438/24 | | 593LP |

were normalized to their maximal values. The calibration procedure to convert the fluorescence changes of a $V_m$ probe into $V_m$ values (mV) has been previously described (*Strünker et al., 2006*; *Seifert et al., 2015*; *Hamzeh et al., 2019*). In brief, sperm were mixed with 2 nM resact at varying extracellular $K^+$ concentrations ($[K^+]_o$). With increasing $[K^+]_o$, the peak amplitude of the resact-evoked hyperpolarization decreased and, eventually, $V_m$ depolarized. A plot of the peak amplitude ($\Delta F/F_0$) versus the $K^+$-Nernst potential for a given $[K^+]_o$ was fitted with a linear fit. The slope of the fitted line yielded the $V_m$ sensitivity (%$\Delta F/F_0$ per mV) of the $V_m$ probe and the x-intercept yielded $V_{rest}$, that is, the $K^+$ Nernst potential at which resact did not change $V_m$. Nernst potentials were calculated assuming an intracellular $K^+$ concentration of 423 mM (*Strünker et al., 2006*). In simultaneous recordings of $[Ca^{2+}]_i$, $pH_i$, and $V_m$, the $V_m$ onset of the $[Ca^{2+}]_i$ and $pH_i$ signals was deduced from their respective latencies.

To determine a probe's crosstalk into orthogonal channels, the fluorescence values recorded in the different channels were plotted against each other for the particular time window indicated in the figure legend and fitted with a linear equation. The slope of the fit was multiplied by 100 to quantify, as a percentage, the extent of crosstalk.

## Caged compounds and flash photolysis

The protocol for loading sperm with BECMCM-cGMP is provided in *Table 1*. For uncaging, sperm were mixed with ASW and allowed to equilibrate in the cuvette for 5–10 s, after which a 50 ms pulse of UV light from a LED (M340L4; Thorlabs) was delivered using a custom-made triggering device.

## Simultaneous ratiometric recording of $[Ca^{2+}]_i$ and $pH_i$ signals in human sperm

Samples of human semen were obtained from volunteers with their prior written consent, under approval of the institutional ethical committees of the medical association Westfalen-Lippe and the medical faculty of the Universtity of Münster (4INie). Human sperm were purified by 'swim-up' into human tubular fluid (HTF) as described previously (*Strünker et al., 2011*). Fura-FF AM (10 µM) was added to a sperm suspension ($10^7$ sperm/ml) supplemented with 0.05% Pluronic F-127 and incubated for 90 min at 37°C. The probe-loaded sperm were pelleted by centrifugation (700 × *g*, 5 min at 37°C), resuspended in HTF, and incubated for 60 min at 37°C to allow de-esterification of intracellular

**Table 3.** Optical configuration for simultaneous ratiometric recording of $[Ca^{2+}]_i$ and $pH_i$ signals in human sperm.

| Fluorescent probe | LED (Thorlabs) | FAST$^M$ modulation frequency (kHz) | Excitation filter (Semrock) | Dichroics | Emission filter (Semrock) |
|---|---|---|---|---|---|
| Fura-FF | M340L4 | 87.3 | 340/22 | HC BS 365 (Semrock) HC BS 409 (Semrock) 470 LPXR (Chroma) | 524/24 |
| | M375L4 | 73.51 | 370/10 | | |
| BCECF | M455L4 | 61.7 | 445/20 | | |
| | M490L4 | 103.7 | 485/20 | | |

FAST$^M$, frequency- and spectrally-tuned multiplexing.

probe. BCECF AM (2 µM) was added to the Fura-FF-loaded sperm and incubated for 5 min; after which, sperm were pelleted (700 × *g*, 5 min at 37°C) and resuspended in HTF. The sperm density was adjusted to $6 \times 10^7$ sperm/ml.

The sperm suspension was rapidly mixed (1:1) with HTF, HTF containing 200 nM progesterone, or HTF containing 60 mM NH$_4$Cl in a microvolume stopped-flow (µSFM, BioLogic) at a flow rate of 1 ml/s, resulting in a dead time of 1.9 ms. The optical configuration is summarized in *Table 3*. Signals to modulate the LEDs were provided by a lock-in amplifier (MFLI, Zurich Instruments) and a waveform generator (Agilent, 33220A), which was synchronized to the 10 MHz clock of the lock-in amplifier. Signals were amplified and frequency filtered by the lock-in amplifier using a third-order (18 dB/octave) lowpass filter and a time constant of 1 ms. Data acquisition was performed as described for sea urchin sperm, but with a sampling rate of 5 ms.

Each signal represents the average of at least three recordings. Dual-excitation ratiometric $[Ca^{2+}]_i$ signals reported by Fura-FF were determined by dividing the fluorescence signal recorded upon excitation at 340 nm over that at 370 nm (340/370). Ratiometric $pH_i$ signals reported by BCECF were determined by dividing the signal recorded upon excitation at 485 nm over that at 445 nm (485/445). Signals are depicted as the percentage change in the ratio with respect to the mean of the first 3–10 data points after mixing ($\Delta R/R_0$ (%)). The baseline $\Delta R/R_0$ obtained upon mixing with HTF alone was subtracted. Signals were also depicted individually for each signal channel as fluorescence changes ($\Delta F/F_0$ (%)) and were calculated as described above for sea urchin sperm.

## Determination of $K_{off}$ for fluorescent $Ca^{2+}$ probes and of signal-to-noise ratios

Calbryte 630 (1 µM), Fluo-4 (1 µM), and Fura-2 (20 µM) dissolved in buffer containing 100 mM KCl, 20 mM HEPES, and 400 µM CaCl$_2$ (pH 7.5) were rapidly mixed (1:1) with buffer containing 10 mM BAPTA (Sigma-Aldrich), 100 mM KCl, and 20 mM HEPES, (pH 7.5) in the µSFM at a flow rate of 1.3 ml/s, resulting in a dead time of 1.5 ms. The ensuing changes in probe fluorescence, reflecting the unbinding of $Ca^{2+}$, were monitored with the optical configuration summarized in *Table 4*.

Signals were amplified and frequency filtered with a third-order (18 dB/octave) lowpass filter and a time constant of 100 µs and were recorded as described for sea urchin sperm, but with a sampling

**Table 4.** Optical configuration for $k_{off}$ determination.

| Fluorescent probe | LED (Thorlabs) | FAST$^M$ modulation frequency (kHz) | Excitation filter (Semrock) | Dichroics | Emission filter (Semrock) |
|---|---|---|---|---|---|
| Fura-2 | M340L4 | 87.31 | 340/22 | HC BS 365 (Semrock) HC BS 409 (Semrock) 525 LPXR (Chroma) | 524/24 |
| | M375L4 | 103.7 | 370/10 | | |
| Fluo-4 | M490L4 | 59.51 | 485/20 | | |
| Calbryte 630 | M565L3 | 47.1 | 586/20 | | 647/57 |

FAST$^M$, frequency- and spectrally-tuned multiplexing.

rate of 100 µs. Each signal represents the average of at least five recordings. Signals are depicted as the relative change in fluorescence ($\Delta F/F_0$) with respect to the baseline signal ($F_0$) recorded in 400 µM $CaCl_2$, that is, in the absence of BAPTA ($F_0$). To determine the $k_{off}$, signal curves were fitted with a monoexponential decay with no fit constraints in GraphPad Prism 9 (Prism). The fluorescence crosstalk between channels was determined as described for recordings from sea urchin sperm.

For evaluation of the S/N ratio, Fura-2 (20 µM) in 100 mM KCl, 20 mM HEPES, and 400 µM $CaCl_2$ (pH 7.5) was mixed with buffer containing 10 mM BAPTA, 100 mM KCl, and 20 mM HEPES (pH 7.5). Fluorescence was excited by light from a single LED (M375L4, fitted with a 370/10 filter) that delivered either continuous or modulated (103.7 kHz) light to the observation cuvette. To detect Fura-2 fluorescence excited by modulated illumination, the recording configuration was the same as described above, except that the time constant of the lock-in amplifier was varied between 100 µs and 2 ms with a matching sample rate. To detect fluorescence excited by continuous illumination, the signal was amplified and filtered through a conventional voltage amplifier (DLPVA-100-B-S: Femto Messtechnik) and subsequently routed to the data acquisition pad (PCI-6221; National Instruments) and Bio-Kine software (BioLogic). The S/N ratio was calculated by dividing the mean signal intensity by the standard deviation over 200 data points.

## Recording of fluorescence spectra

Fluorescence spectra were recorded from 50 µl of either a probe-containing solution or a probe-loaded sperm suspension in 384-well plates (Greiner) with a fluorescent plate reader (CLARIOStar, BMG) in spectral scanning mode using bottom optics.

For sea urchin sperm, dry sperm was diluted 1:6 (v/v) in ASW and loaded with a fluorescent probe according to *Table 1*. The probe-loaded sperm suspension was further diluted 1:10 (v/v) with ASW for spectral acquisition.

For human sperm, loading with either Fura-FF AM or BCECF AM was performed as described above. The probe-loaded sperm suspension ($10^7$ sperm/ml) was centrifuged (700 × *g* for 5 min at 37°C) and resuspended in different buffers to a concentration of $2 \times 10^7$ sperm/ml. Fura-FF-loaded sperm were resuspended in HTF supplemented with 10 µM ionomycin (Biomol) or $Ca^{2+}$-free HTF supplemented with 5 mM EGTA (Sigma-Aldrich). BCECF-loaded sperm were resuspended in HTF adjusted to either pH 6.5 or pH 8.5.

For spectral acquisition of probes in solution, either Fura-2, Fluo-4, or Calbryte 630 was diluted to 1 µM in a buffer containing 100 mM KCl, 20 mM HEPES (pH 7.5), and either 400 µM $CaCl_2$ or 5 mM EGTA.

## Single-cell recordings of intracellular cAMP and $[Ca^{2+}]_i$

HEK293 cells (flp-In-293) stably transfected with expression constructs encoding the biogenic amine receptor DmOCTβ1R from *Drosophila melanogaster* (**Balfanz et al., 2005**) and the bovine CNGA2-TM channel (**Wachten et al., 2006**; **Schröder-Lang et al., 2007**) were cultured in DMEM plus GlutaMAX (Thermo Fisher Scientific) supplemented with 10% fetal bovine serum (Biochrom) and 1× penicillin/streptomycin (Thermo Fisher Scientific) with selective pressure provided by G418 (800 µg/ml) (Thermo Scientific) and hygromycin (100 µg/ml) (Thermo Fisher Scientific) for constitutive expression of DmOCTB1R, respectively. Authentication was performed by functional tests as shown in *Figure 10*. Cell lines were negatively tested for mycoplasma contamination. The cells were seeded onto 5 mm glass coverslips (#1; Thermo Fisher Scientific) that were coated with poly-L-lysine (Sigma-Aldrich). To yield cells expressing a FRET-based cAMP biosensor, DmOCTβ1 receptors, and CNGA2 channels, cells at 50–60% confluency were transfected using Lipofectamine 2000 (Invitrogen) according to the manufacturer's protocol with either pc3.1-mlCNBD-FRET (high cAMP affinity), pc3.1-mlCNBD-FRET-R307Q (non-binding) (**Mukherjee et al., 2016**), or pc3.1-mlCNBD-FRET-M329C (low cAMP affinity). Using pc3.1-mlCNBD-FRET as a template, we performed QuikExchange (Agilent) to introduce the M329C mutation. All vectors are based on the pcDNA3.1(+) vector and contain a neomycin-resistant cassette for selection in mammalian cells. To load the cells with Calbryte 630 AM, cells adhered to cover slips were washed once with ES (in mM): 120 NaCl, 5 KCl, 2 $CaCl_2$, 10 HEPES, and 10 glucose, adjusted to pH 7.5 with NaOH and then incubated in ES supplemented with 10 µM Calbryte 630 AM, 0.05% Pluronic, and 3 mM Probenecid (Sigma-Aldrich) for 10 min at 37°C. Cells were washed once with ES to remove unloaded probe. Coverslips were transferred to a custom-built headstage

chamber that was fit with a custom-built gravity flow-perfusion system and imaged with an inverted microscope (IX73; Olympus). The excitation module consisted of a blue LED (M455L4; Thorlabs) fitted with a 438/24 nm filter (Semrock) and a green LED (M565L3; Thorlabs) fitted with a 565/24 nm filter (Semrock). The blue and green LEDs were modulated at 36.1 and 49.5 kHz, respectively, by lock-in amplifiers (MFLI, Zurich Instruments, or SR844 RF, Stanford Research Systems). The modulated output of these LEDs was combined on a dichroic (470 LPXR; Chroma), passed through a neutral density filter (NDUV20A; Thorlabs) and a CFP-YFP-mCherry filter cube (AHF Analysetechnik), and focused onto the sample with a ×60 water immersion objective (UPlanSApo, numerical aperture [NA] 1.2; Olympus). The modulated fluorescence signals were directed through the CFP-YFP-mCherry filter cube and split by a dichroic (525 LPXR; Chroma) onto two PMTs. One PMT was fitted with a 475/28 nm bandpass filter (Semrock) to collect fluorescence from the FRET donor (cerulean). The other PMT was fitted with a 578/105 nm bandpass filter (Semrock) to collect fluorescence from the FRET acceptor (citrine) and the $Ca^{2+}$ probe (Calbryte 630). Signals were routed to lock-in amplifiers, where they were amplified and frequency filtered with a third-order (18 dB/octave) lowpass filter and a time constant of 10 ms. Data were acquired at 200 Hz with an analog-to-digital converter (Axon Digidata 1550A; Molecular Devices) and pCLAMP software (Molecular Devices). Selection of single cells was performed with brightfield illumination using a halogen light source (TH4-200; Olympus) and a condenser (IX2-MLWPO, NA 0.5; Olympus). An aluminum mirror (Chroma) was temporarily installed in the optical path to divert transmitted light to a camera (IDS Imaging). Cells expressing the FRET-based cAMP sensor were selected based on fluorescence. If necessary, an aperture was adjusted to encircle the cell of interest to isolate its fluorescence from surrounding cells. After selection, the mirror was removed, directing the fluorescence to the PMTs. To measure cAMP and $[Ca^{2+}]_i$ signals, cells were perfused with ES for 20 s, then perfusion was switched to ES containing 20 µM octopamine. The FRET ratio, that is, cAMP signal, was calculated by dividing the signal recorded in the donor (cerulean) channel by the signal recorded in the acceptor (citrine) channel. The FRET ratio is shown as the percent change in the FRET ratio with respect to the mean value of the first 1 s of recording ($\Delta R/R_0$ (%)). The $[Ca^{2+}]_i$ signals are shown as the percent change in fluorescence with respect to the mean of the first 1 s of recording ($\Delta F/F_0$). The latencies of the FRET and $[Ca^{2+}]_i$ signals were deduced from the signal time course.

## Acknowledgements

We thank all present and former members of U Benjamin Kaupp's lab for continuously collecting, handling, and shipping of sea urchin sperm to support this project. This work was supported by the German Research Foundation (STR 1342/3-1 to TS, CRU326 to CB and TS, GRK2515 to TS), under Germany's Excellence Strategy – EXC2151 – 390873048 (to DW), the 'Innovative Medical Research' of the University of Münster Medical School (BR 121507 to CB), the Center for Clinical Research, Münster (IZKF; Str/014/21 to TS), and the NIH NIGMS (R35GM119855 to EWM).

## Additional information

### Funding

| Funder | Grant reference number | Author |
|---|---|---|
| Deutsche Forschungsgemeinschaft | STR 1342/3-1 | Timo Strünker |
| Deutsche Forschungsgemeinschaft | CRU326 | Timo Strünker Christoph Brenker |
| Deutsche Forschungsgemeinschaft | EXC2151 - 390873048 | Dagmar Wachten |
| Innovative Medical Research of the University of Muenster Medical School | BR 1 2 15 07 | Christoph Brenker |
| Center for Clinical Research, Münster | Str/014/21 | Timo Strünker |

| Funder | Grant reference number | Author |
| --- | --- | --- |
| National Institute of General Medical Sciences | R35GM119855 | Evan W Miller |
| Deutsche Forschungsgemeinschaft | GRK2515 | Timo Strünker |

The funders had no role in study design, data collection and interpretation, or the decision to submit the work for publication.

## Author contributions

Michelina Kierzek, Formal analysis, Investigation, Methodology, Visualization, Writing – original draft, Writing – review and editing; Parker E Deal, Evan W Miller, Dagmar Wachten, Arnd Baumann, Resources, Writing – review and editing; Shatanik Mukherjee, Investigation, Resources, Writing – review and editing; U Benjamin Kaupp, Resources, Writing – original draft, Writing – review and editing; Timo Strünker, Christoph Brenker, Conceptualization, Formal analysis, Funding acquisition, Investigation, Methodology, Project administration, Supervision, Visualization, Writing – original draft, Writing – review and editing

## Author ORCIDs

Evan W Miller http://orcid.org/0000-0002-6556-7679
Shatanik Mukherjee http://orcid.org/0000-0002-7359-9339
Dagmar Wachten http://orcid.org/0000-0003-4800-6332
Timo Strünker http://orcid.org/0000-0003-0812-1547
Christoph Brenker http://orcid.org/0000-0002-4230-2571

## Decision letter and Author response

Decision letter https://doi.org/10.7554/eLife.63129.sa1
Author response https://doi.org/10.7554/eLife.63129.sa2

# Additional files

## Supplementary files

• Transparent reporting form

## Data availability

All data generated or analysed during this study are included in the manuscript and supporting files. Source Data files have been provided for Figures 3, 5, 7, 9 and 11.

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
