## [Editor Report]

The number and temporal resolution of fluorescent probes that can be used simultaneously to interrogate cell signaling pathways are currently limited by overlap of their excitation and emission spectra. This study introduces a new technique to overcome these limitations and enable simultaneous recording of multiple probes at millisecond time resolution. The technique will facilitate studies of the temporal and causal relationships among many signaling pathways that can be targeted with fluorescent probes.

---

## [Decision Letter]

**Decision letter after peer review:**

Thank you for submitting your article "Simultaneous recording of multiple signaling events by frequency-and spectrally-tuned multiplexing of fluorescent probes" for consideration by *eLife*. Your article has been reviewed by 3 peer reviewers, one of whom is member of our Board of Reviewing Editors, and the evaluation has been overseen by Kenton Swartz as the Senior Editor. The reviewers have opted to remain anonymous.

The reviewers have discussed the reviews with one another and the Reviewing Editor has drafted this decision to help you prepare a revised submission.

As the editors have judged that your manuscript is of interest, but as described below that additional experiments are required before it is published, we would like to draw your attention to changes in our revision policy that we have made in response to COVID-19 (https://elifesciences.org/articles/57162). First, because many researchers have temporarily lost access to the labs, we will give authors as much time as they need to submit revised manuscripts. We are also offering, if you choose, to post the manuscript to bioRxiv (if it is not already there) along with this decision letter and a formal designation that the manuscript is "in revision at eLife". Please let us know if you would like to pursue this option. (If your work is more suitable for medRxiv, you will need to post the preprint yourself, as the mechanisms for us to do so are still in development.)

Summary:

Simultaneous recording of multiple signals in living cells is a powerful approach for determining relationships among events in biological signal transduction. A current limitation results from emission spectral overlap of fluorescence-based probes, which limits the number of signals that can be monitored and the temporal resolution of the recordings. This paper introduces a technique, FASTM, which uses phase-sensitive signal detection to overcome these limitations. Examples include simultaneous recordings of Ca^2+^, Na^+^, pH and V_m_ (in combinations of three) with millisecond resolution, and Ca^2+^ and cAMP using a FRET-based cAMP biosensor in single cells. These applications demonstrate the potential of FAST^M^ to simultaneously record and distinguish multiple biological signals at high temporal resolution, using probes with distinct excitation spectra but overlapping emission spectra.

Essential revisions:

The reviewers agreed that the experimental work was carried out well and adequately described, and that the FAST^M^ technique may offer a useful tool for multi-probe measurements. However, the criterion for acceptance as a Tools and Resources papers in *eLife* is that the new technique should enable experiments leading to new biological insights or mechanisms. In this regard, a weakness of the paper is that the critical advantages of FAST^M^ over existing approaches are not convincingly or explicitly demonstrated. This issue must be addressed along with others described below.

1. FAST^M^ is an effective way of separately measuring the overlapping emissions from probes that have separable excitation spectra. However, rapidly switchable excitation filter wheels or galvanometer-based devices also solve this problem and are widely used. Please describe specific experimental applications where FAST^M^ provides advantages over current approaches. Be specific about how the characteristics of FAST^M^ will enable measurements that cannot be made with existing methods, and why this is important. This is needed to demonstrate that FAST^M^ will enable new kinds of measurements not possible with current methods.

2. One important question is how signal demodulation affects the signal-to-noise ratio and temporal resolution as compared with the standard excitation switching technique. A quantitative comparison of S/N ratio with and without FAST^M^ could be made for cell suspensions and also in single cells (where the effects may be more dramatic).

3. To bolster the innovative aspect and new applications for this technique, please show an example of ratiometric Ca^2+^ or pH measurements (e.g., fura-2).

4. (l. 124-126) RhoVR can be monitored separately by its red- shifted excitation/emission (Figure 3). The question is whether FAST^M^ confers any advantage here. One possible advantage of FAST^M^ could be to reduce photobleaching, because more of the emission band could be collected even though it overlaps with emissions of other probes. An example of this should be shown.

5. The required conditions for application of FAST^M^ should be more clearly stated, and much earlier in the paper (before the Discussion). For example, the excitation spectra of two probes with overlapping emissions need to differ but need not be entirely distinct, as long as one of the probes can be uniquely excited (e.g., see Figure 6A, V_m_ and Ca^2+^). It may be useful to use a diagram to illustrate the various spectral situations for which FAST^M^ would be advantageous.

6. Figure 9 is confusing. The examples in A show that Ca^2+^ signal latency increases with the type of cAMP sensor, yet the FRET cAMP signal doesn't appear to be much affected. The sensor indicates free cAMP, which appears similar with WT sensor or M329C sensor, yet the Ca^2+^ signal latency is very different. Why? Also, how is a FRET peak measured in these responses when the FRET appears to simply rise to a plateau? The results here don't appear to support the conclusions.

7. Please comment on why the depolarizing signal that results from uncaging cGMP is significantly larger than when stimulating with resact.

8. (l. 266-273, l. 296): Much of the significance of the FAST^M^ technique rests on claims for additional probes and combinations that could be used. Some examples of other combinations should be supplied, perhaps with >3 probes that could be distinguished by FAST^M^. Fluorescence and absorbance probes? Multiple FRET probes? A table may be an effective way to display these examples.

9. l. 61-64 states that multiplexing of signaling events has been limited to simultaneous recording of two ion species or one ion and Vm. This is not strictly true – there are several papers in the sperm field where cytometry and imaging cytometry allow simultaneous measurements of three or more dyes, and several methods already exist for disentangling multiple signals using phase differences and modulation (see references below).

Loew LM. Voltage-sensitive dyes: measurement of membrane potentials induced by DC and AC electric fields. Bioelectromagnetics. 1992;Suppl 1:179-89.

Carlsson K et al., Simultaneous confocal recording of multiple fluorescent labels with improved channel separation. J Microsc. 1994 176:287-99.

Ronzitti E, et al., Frequency dependent detection in a STED microscope using modulated excitation light. Opt Express. 2013 21:210-9.

Berndt KW, et al., A 4-GHz frequency-domain fluorometer with internal microchannel plate photomultiplier cross- correlation. Anal Biochem. 1991 192:131-7.

Wei-LaPierre L, et al., Respective contribution of mitochondrial superoxide and pH to mitochondria-targeted circularly permuted yellow fluorescent protein (mt-cpYFP) flash activity. J Biol Chem. 2013 288:10567-77.

Wu F, et al., Frequency division multiplexed multichannel high-speed fluorescence confocal microscope. Biophys J. 2006 91:2290-6.

Schreiber U. Detection of rapid induction kinetics with a new type of high- frequency modulated chlorophyll fluorometer. Photosynth Res. 1986 9:261-72.

Kalaji HM, et al., Experimental in vivo measurements of light emission in plants: a perspective dedicated to David Walker. Photosynth Res. 2012 114:69-96.

---

## [Author Response]

Essential revisionsThe reviewers agreed that the experimental work was carried out well and adequately described, and that the FAST^M^ technique may offer a useful tool for multi-probe measurements. However, the criterion for acceptance as a Tools and Resources papers in eLife is that the new technique should enable experiments leading to new biological insights or mechanisms. In this regard, a weakness of the paper is that the critical advantages of FAST^M^ over existing approaches are not convincingly or explicitly demonstrated. This issue must be addressed along with others described below.1. FAST^M^ is an effective way of separately measuring the overlapping emissions from probes that have separable excitation spectra. However, rapidly switchable excitation filter wheels or galvanometer-based devices also solve this problem and are widely used. Please describe specific experimental applications where FAST^M^ provides advantages over current approaches. Be specific about how the characteristics of FAST^M^ will enable measurements that cannot be made with existing methods, and why this is important. This is needed to demonstrate that FAST^M^ will enable new kinds of measurements not possible with current methods.

We now outline more clearly the rationale for developing FAST^M^ and its advantages over existing techniques. In a nutshell, FAST^M^ greatly expands the application range of time-resolved multiplexing, because it enables discrimination of simultaneously recorded probes based on their excitation, emission, or both. Thereby, FAST^M^ overcomes the fluorescence crosstalk that has, thus far, prevented multiplexing of more than two probes with *millisecond* time resolution that is required to resolve fast biochemical, electrical, and ionic signaling events. Therefore, we envisage that FAST^M^ will be widely adopted for multiplexing of rapid signals in aqueous solutions, single cells, and cell populations.

As an example, the temporal resolution of FAST^M^ is independent of the number of probes and only limited by the time constants of the lock-in amplifiers and/or detectors, which are generally in the range of microseconds. To demonstrate FAST^M^´s exceptional temporal resolution, we simultaneously monitored the kinetics of Ca^2+^ dissociation from Fura-2 (dual-excitation recording), Fluo-4, and Calbryte 630 by recording four fluorescence channels each with a time constant of 100 µs (Figure 9). By contrast to FAST^M^, the temporal resolution of multiplexing based on excitation-switching is set by the switching time and, therefore, decreases inevitably with the number of probes. Using state-of-the-art filter wheels (e.g., Chroma-OptoSpin4, ASI FW-1000 High Speed Filter Wheel, or Thorlabs filter wheel), multiplexing of four fluorescence channels could be performed with a time resolution of > 150 ms; galvanometerbased devices (e.g., DeltaRAM X Random Access Monochromator, PTI/Horiba) achieve a time resolution of > 20 ms. The time resolution of these excitation-switching approaches would, thus, be insufficient to resolve such rapid signals.

Concerning the comparison with present multiplexing techniques, see also the discussion and response to comment #2.

2. One important question is how signal demodulation affects the signal-to-noise ratio and temporal resolution as compared with the standard excitation switching technique. A quantitative comparison of S/N ratio with and without FAST^M^ could be made for cell suspensions and also in single cells (where the effects may be more dramatic).

We performed additional experiments to address these two important questions. Signal demodulation does not impair the signal-to-noise (S/N) ratio. In fact, the hallmark of phase-sensitive signal detection is that it enhances rather than decreases the S/N (e.g., Meade 1983), independent of whether fluorescent probes are recorded in solution, cell suspensions, or single cells. We now compare recordings performed with or without FAST^M^ to bolster this common observation with our own data (Figure 9—figure supplement 2). Moreover, demodulation does not affect the time resolution; please see the response to comment #1.

In summary, compared to the excitation-switching techniques, FAST^M^ features superior temporal resolution and sensitivity.

3. To bolster the innovative aspect and new applications for this technique, please show an example of ratiometric Ca^2+^ or pH measurements (e.g., fura-2).

We now show in Figure 9 simultaneous, ratiometric recordings of Fura-2 and two non-ratiometric probes (see response to comment #2). In addition, we simultaneously recorded rapid [Ca^2+^]_i_ and pH_i_ signals in human sperm by simultaneous ratiometric recording of both Fura-FF and BCECF (summarized in Figure 7).

4. (l. 124-126) RhoVR can be monitored separately by its red- shifted excitation/emission (Figure 3). The question is whether FAST^M^ confers any advantage here. One possible advantage of FAST^M^ could be to reduce photobleaching, because more of the emission band could be collected even though it overlaps with emissions of other probes. An example of this should be shown.

There seems to be a misunderstanding. There is no RhoVR signal in the Fura-2 and the BCECF channel with or without FAST^M^. However, without FAST^M^, Fura-2 and BCECF signals are detected in the RhoVR channel (Figure 3). Therefore, simultaneous recording of RhoVR, Fura-2, and BCECF requires FAST^M^. We rephrased the text to emphasize that aspect.

Indeed, FAST^M^ reduces photobleaching because reasonable S/N ratios can be reached with lower excitation-light intensity (see response to comment #3). This is now mentioned in the text. We did not perform experiments to bolster this claim with data, because the relation of excitation-light intensity, bleaching, and S/N ratio is well known.

5. The required conditions for application of FAST^M^ should be more clearly stated, and much earlier in the paper (before the Discussion). For example, the excitation spectra of two probes with overlapping emissions need to differ but need not be entirely distinct, as long as one of the probes can be uniquely excited (e.g., see Figure 6A, V_m_ and Ca^2+^). It may be useful to use a diagram to illustrate the various spectral situations for which FAST^M^ would be advantageous.

We agree and amended the text accordingly. Moreover, we now show a schematic in Figure 1 that illustrates the spectral requirements for probes to be combined for FAST^M^. Possible spectral set-ups for FAST^M^ are also illustrated by the various different experiments that were performed.

6. Figure 9 is confusing. The examples in A show that Ca^2+^ signal latency increases with the type of cAMP sensor, yet the FRET cAMP signal doesn't appear to be much affected. The sensor indicates free cAMP, which appears similar with WT sensor or M329C sensor, yet the Ca^2+^ signal latency is very different. Why? Also, how is a FRET peak measured in these responses when the FRET appears to simply rise to a plateau? The results here don't appear to support the conclusions.

We apologize for the confusing wording. We amended the figures and rephrased the text to be clearer. In general, based on this set of experiments, we can conclude that FAST^M^ allows simultaneous recording of probes in single cells. Moreover, we demonstrate that introduction of the cAMP biosensor delays activation of the downstream effector CNGA2, which seems to get its share of cAMP only after the, compared to CNGA2, high-affinity cAMP sensor is saturated with cAMP. This example illustrates how probe-related perturbations of signaling pathways can be unveiled by multiplexing experiments. We do not know why the time to saturation of the cAMP signal and, thereby, also the Ca^2+^-signal latency is prolonged when expressing the WT versus mutant sensor. This might be due to differences in expression levels, which we did not analyze.

7. Please comment on why the depolarizing signal that results from uncaging cGMP is significantly larger than when stimulating with resact.

The amplitude of the depolarization features a bell-shaped dose dependence (Strünker et al., 2006, *Nature Cell Biology*) and varies slightly among sperm samples and experiments. We do not know whether the differences in the extent of the depolarization upon cGMP uncaging vs. resact stimulation arise due to biological variation, different stimulus strength, or a combination of both. This would have required investigating resact- and cGMP-evoked responses in sperm from the same sperm sample sideby-side over a broad range of concentrations and light intensities, respectively. In the present study, we did not perform such systematic comparisons, because the aim was rather to test whether FAST^M^ is compatible with flash photolysis of caged compounds.

8. (l. 266-273, l. 296): Much of the significance of the FAST^M^ technique rests on claims for additional probes and combinations that could be used. Some examples of other combinations should be supplied, perhaps with >3 probes that could be distinguished by FAST^M^. Fluorescence and absorbance probes? Multiple FRET probes? A table may be an effective way to display these examples.

In the revised manuscript, we now demonstrate the broad applicability of FAST^M^ for simultaneous recording of probes in three different cell types and in solution using eight different combinations of twelve different fluorophores, covering the entire visible-light spectrum. We now also provide examples with multiplexing of four fluorescence channels and even two ratiometric probes at the same time. Moreover, in the material and methods section, we provide a table with the particular settings and conditions used for the various combinations of probes; and we now show a schematic illustrating the spectral requirements for probes to be combined for FAST^M^ (Figure 1). We argue that this not only showcases the versatility and wide applicability of FAST^M^, but also provides the requisite know-how to both adopt the technique for other applications and explore further combinations of probes. A table providing even more examples can only be incomplete, considering the multitude of available fluorophores and possible combinations thereof. Therefore, we refrain from providing such a table, unless the referees insist.

9. l. 61-64 states that multiplexing of signaling events has been limited to simultaneous recording of two ion species or one ion and V_m_. This is not strictly true – there are several papers in the sperm field where cytometry and imaging cytometry allow simultaneous measurements of three or more dyes, and several methods already exist for disentangling multiple signals using phase differences and modulation (see references below).

Indeed, flow- and imaging-cytometry allows for multiplexing of several probes in sperm and somatic cells, but only with a time resolution in the range of several seconds. Multiplexing with *millisecond* time resolution is not feasible with such techniques. Therefore, we maintain the statement that FAST^M^ is the first technique that enables multiplexing of three or more *rapid* signaling events occurring on a *millisecond or even sub-millisecond* time scale.

Furthermore, the references listed by the referees are interesting examples for the use of the lock-in technique and phase-sensitive detection in biomedical research; that is why we referred to some of these studies in the introduction of the original and revised manuscript. However, none of these studies demonstrated or suggested its application for *rapid*, *time-resolved* multiplexing of probes like in FAST^M^.

In the following, we briefly comment on each of these studies.

Loew LM. Voltage-sensitive dyes: measurement of membrane potentials induced by DC and AC electric fields. Bioelectromagnetics. 1992;Suppl 1:179-89.

This review highlights the use of voltage-sensitive dyes to monitor V_m_. Some of the cited studies use phase-sensitive signal detection to selectively isolate signals originating from AC fields, which is a prime example for the use of this approach to enhance the S/N ratio, allowing recovering small signals on a noisy background. There is, however, no link to the use of this approach for multiplexing.

Carlsson K et al., Simultaneous confocal recording of multiple fluorescent labels with improved channel separation. J Microsc. 1994 176:287-99.

This study describes the use of phase-sensitive signal detection in a confocal microscope to reduce cross-talk between different fluorescence channels. In contrast to our approach, the focus is on the separation of fluorescence signals that are steady rather than changing. Nevertheless, we thank the reviewer for highlighting this study and now also refer to it in the introduction.

Ronzitti E, et al., Frequency dependent detection in a STED microscope using modulated excitation light. Opt Express. 2013 21:210-9.

Here, modulation of the excitation light was used to improve the versatility of STED microscopes, i.e., to separate the excitation and STED laser, which increases the spatial resolution. There is, however, no link to multiplexing.

Berndt KW, et al., A 4-GHz frequency-domain fluorometer with internal microchannel plate photomultiplier cross- correlation. Anal Biochem. 1991 192:131-7.

This study describes the development and application of a PMT for frequency-domain lifetime and anisotropy measurements. Even though frequency-domain lifetime measurements and FAST^M^ rely on modulated light sources, the purpose of the approach is entirely different.

Wei-LaPierre L, et al., Respective contribution of mitochondrial superoxide and pH to mitochondria-targeted circularly permuted yellow fluorescent protein (mt-cpYFP) flash activity. J Biol Chem. 2013 288:10567-77.

This study rather highlights the limitations of quasi-simultaneous recording to multiplex genetically encoded fluorophores and pH-sensitive probes. This approach achieved a temporal resolution of only ≥ 1 s.

Wu F, et al., Frequency division multiplexed multichannel high-speed fluorescence confocal microscope. Biophys J. 2006 91:2290-6.

This study provides another interesting example of the use of phase-sensitive signal detection. The authors excite different spots in the sample with light modulated at different frequencies and, thereby, measure the fluorescence from different spots with a single detector. This study is unrelated to fluorescence multiplexing and, thus, FAST^M^, but provides the blueprint for future studies using singlecell FAST^M^ with spatial resolution.

Schreiber U. Detection of rapid induction kinetics with a new type of high- frequency modulated chlorophyll fluorometer. Photosynth Res. 1986 9:261-72.

This is another example for the use of phase-sensitive signal detection to enhance the S/N ratio. There is no link to multiplexing.

Kalaji HM, et al., Experimental in vivo measurements of light emission in plants: a perspective dedicated to David Walker. Photosynth Res. 2012 114:69-96.

This review tackles many diverse optical devices to study photosynthesis and chlorophyll fluorescence, but as already mentioned above, nothing is directly related to the principles underlying multiplexing using FAST^M^.